# Selective enrichment of plasma cell-free messenger RNA in cancer-associated extracellular vesicles

Hyun Ji Kim [1,2], Matthew J. Rames [1,2], Florian Goncalves[1,2], C. Ward Kirschbaum[1],
Breeshey Roskams-Hieter[1], Elias Spiliotopoulos[1], Josephine Briand[1], Aaron Doe[1], Joseph Estabrook[1,3],
Josiah T. Wagner[1,4], Emek Demir[1,3,5], Gordon Mills [6] & Thuy T. M. Ngo [1,2,5,6 ✉]

Extracellular vesicles (EVs) have been shown as key mediators of extracellular small RNA transport. However, carriers of cell-free messenger RNA (cf-mRNA) in human biofluids and their association with cancer remain poorly understood. Here, we performed a transcriptomic analysis of size-fractionated plasma from lung cancer, liver cancer, multiple myeloma, and healthy donors. Morphology and size distribution analysis showed the successful separation of large and medium particles from other soluble plasma protein fractions. We developed a strategy to purify and sequence ultra-low amounts of cf-mRNA from particle and protein enriched subpopulations with the implementation of RNA spike-ins to control for technical variability and to normalize for intrinsic drastic differences in cf-mRNA amount carried in each plasma fraction. We found that the majority of cf-mRNA was enriched and protected in EVs with remarkable stability in RNase-rich environments. We observed specific enrichment patterns of cancer-associated cf-mRNA in each particle and protein enriched subpopulation. The EV-enriched differentiating genes were associated with specific biological pathways, such as immune systems, liver function, and toxic substance regulation in lung cancer, liver cancer, and multiple myeloma, respectively. Our results suggest that dissecting the complexity of EV subpopulations illuminates their biological significance and offers a promising liquid biopsy approach.

---

[1] Cancer Early Detection Advanced Research Center (CEDAR), Knight Cancer Institute, Oregon Health and Science University, Portland, OR, USA.
[2] Department of Biomedical Engineering, Oregon Health and Science University, Portland, OR, USA. [3] Computational Biology Program, Oregon Health and Science University, Portland, OR, USA. [4] Molecular Genomics Laboratory, Providence Health and Services, Portland, OR, USA. [5] Department of Molecular and Medical Genetics, Oregon Health and Science University, Portland, OR, USA. [6] Division of Oncological Sciences, Knight Cancer Institute, Oregon Health and Science University, Portland, OR, USA. ✉email: ngth@ohsu.edu

Cell-free RNA (cfRNA), also known as extracellular RNA (exRNA), is present in many bodily fluids, including cerebrospinal fluid (CSF), saliva, serum, urine, and plasma[1–3]. The most commonly reported cfRNA biotypes are small non-coding RNAs (ncRNAs), which include microRNAs (miRNAs), transfer RNA (tRNA) fragments, and piwi-interacting RNAs (piRNAs)[4]. Studies have also found messenger RNAs (mRNAs) and long noncoding RNAs (lncRNAs) in biofluids[5–9]. While cfRNA has been considered more fragile and less abundant than cell-free DNA because of high RNase levels in blood[10], several studies have demonstrated that miRNAs present in blood are remarkably stable[11,12]. Many cfRNA studies focused on characterizing cf-miRNA due to its stability in bodily fluids[11,12]. However, we and others have recently demonstrated that cf-mRNAs also can carry stable and disease-specific signatures that can be exploited as non-invasive biomarkers[8,13–18].

cfRNAs are associated with multiple subclasses of carriers including extracellular vesicles (EVs), ribonucleoproteins, and also lipoproteins such as: chylomicrons, very low-density lipoprotein (VLDL), low-density lipoprotein (LDL) and high-density lipoprotein (HDL). The types and proportions of RNA cargos packaged in different carriers may depend on the biofluid type, pre-analytical factors, and physiological or pathological conditions. The NIH Extracellular RNA (exRNA) Communication Consortium created an exRNA atlas resource that extensively compared major carriers of miRNA through computational deconvolution across 19 studies[2,19]. These comparative statistical studies identified five of the miRNA cargo types to be associated with known vesicular and non-vesicular carriers[2]. While these studies implicated EVs as a key mediator of RNA transport, others have shown RNA binding protein complexes can carry miRNA independent of vesicles in human plasma[12,13,20]. Identifying cf-mRNA carriers that provide stability in RNase-rich plasma may reveal basic biology and function, as well as a potential distinct form of carrier-specific biomarkers for clinical applications. In conditioned cell-culture media, cf-mRNA was shown to be associated with EVs[13,20]. However, which carriers associate with cf-mRNA in human biofluid remains poorly characterized[1,21].

One of the potential cf-mRNA carriers suggested by previous studies using cell-culture models and extracellular long RNA sequencing is EVs[13,20,21], which are a collective term for various vesicles that are distinguished based on their size and biogenesis[22]. The discovery of EVs containing cfRNA has sparked considerable interest in understanding the role of these vesicles in intercellular communication and potential clinical applications[13,23,24]. EVs with a diameter larger than 100 nm are categorized as medium EVs, while EVs smaller than 100 nm are categorized as small EVs[25]. Prior research demonstrates that EVs harbor various molecular cargoes, including nucleic acids and proteins associated with their cells of origin, which provides the potential for both diagnostics and prognostics[9,26–28]. In particular, the miRNA content of EVs in plasma across clinical human samples varied remarkably[29,30]. However, little is known regarding whether EVs or protein complexes are the major carrier of cf-mRNA in complex biofluids such as plasma, and whether circulating cf-mRNA associated with EVs can distinguish cancer patients from healthy controls.

Cell culture studies have begun to determine how oncogenic signaling directs extracellular miRNA into EVs[13,31,32]. Specific oncogenes increase the abundance of miRNA and alter EV miRNA composition[31]. These exosomal miRNAs can affect recipient cell phenotypes by altering gene expression and cellular functions[28]. Recent research shows that tumor-derived EVs can selectively package miRNA contents[28,32,33]. In the context of cancer progression, mutations on oncogenes such as KRAS influence the selective packaging of genetic materials into vesicles in cell culture media[28]. Mutated KRAS decreased Argonaut 2 (Ago2) association with different miRNAs being selectively sorted into EVs. Despite evidence for selective miRNA packaging in cancer model cell systems, studies of the effects of disease and genetic aberrations on mRNA content of EVs from human plasma remains under-explored.

In this study, we aimed to identify cf-mRNA carriers in human plasma and to determine whether the abundance or identity of cf-mRNA is altered in different cancers. We developed a protocol to fractionate extracellular carriers in plasma into large and medium particles from other soluble plasma protein fractions using size-exclusion chromatography (SEC). We characterized biophysical properties of extracellular fractions using transmission electron microscopy (TEM) and tunable resistive pulse sensing technology as orthogonal approaches. We then sequenced cf-mRNA from size-fractionated plasma. We incorporated synthetic RNA spike-ins to control for technical variability and intrinsic differences in the amount of cf-mRNA carried in each plasma fraction. Using this approach, we performed transcriptomic analysis of 120 size-fractionated plasma samples from healthy participants and those with lung cancer, liver cancer and multiple myeloma. We also performed immunoprecipitation followed by western blotting and qRT-PCR to further confirm EVs are the major carriers of cf-mRNA in plasma. Furthermore, RNase treatment assays revealed cf-mRNA are protected in EVs. In addition, we observed specific enrichment patterns of cf-mRNA with associated biological pathways in vesicular and non-vesicular subpopulations from different cancer samples.

## Results

**Characterization of human plasma size fractionation.** We investigated potential vesicular and non-vesicular carriers of cf-mRNA in plasma by size fractionation using size exclusion chromatography (SEC). We fractionated the plasma using SEC, and collected a series of 2 mL fractions: 0–2 mL (FR2), 2–4 mL (FR4), 4–6 mL (FR6), etc. (Fig. 1a). Fractions between 2 and 12 mL predominantly contained particles approximately 50–300 nm in diameter as measured by tunable resistive pulse sensing technology (qNano) (Fig. 1b). Fractions collected from the 14 mL elution volume and beyond were below the particle size detection limit ( < 50 nm) of qNano. Absorbance at 280 nm showed that the majority of plasma proteins were eluted beyond the 12 mL elution volume (Fig. 1b). To accurately characterize the distribution of heterogeneous particle sizes, we analyzed their morphology in individual fractions collected from 2 mL to 12 mL using transmission electron microscopy (TEM) (Fig. 1c & Supplementary Fig. 1). The size histogram measured by TEM of the particles eluted between 2 and 12 mL displayed three main distribution ranges: larger than 100 nm, between 50 and 100 nm, and less than 50 nm with the dominant peak around 30 nm (Supplementary Fig. 2). The majority of particles larger than 100 nm were eluted in the first 4 mL fraction (FR4) while particles between 50 and 100 nm were enriched in the next 4–8 mL (FR6 and FR8) (Fig. 1d and Supplementary Data 1). Fractions from 8 to 12 mL (FR10 and FR12) were dominated by less than 50 nm particles (Fig. 1d and Supplementary Data 1). Therefore, we divided fractionated plasma into 4 mL fractions by similar particle sizes potentially containing both EVs and other lipoprotein particles. We referred individual fraction as large particles (FR14: eluted from 0 to 4 mL), medium particles (FR58: eluted from 4 to 8 mL), and small particles (FR912: eluted from 8 to 12 mL) to analyze their cf-mRNA content (Fig. 1e). We also divided protein-enriched fractions of plasma into early-eluting protein fractions (FR1619: eluted from 15 to 19 mL), the middle-eluting

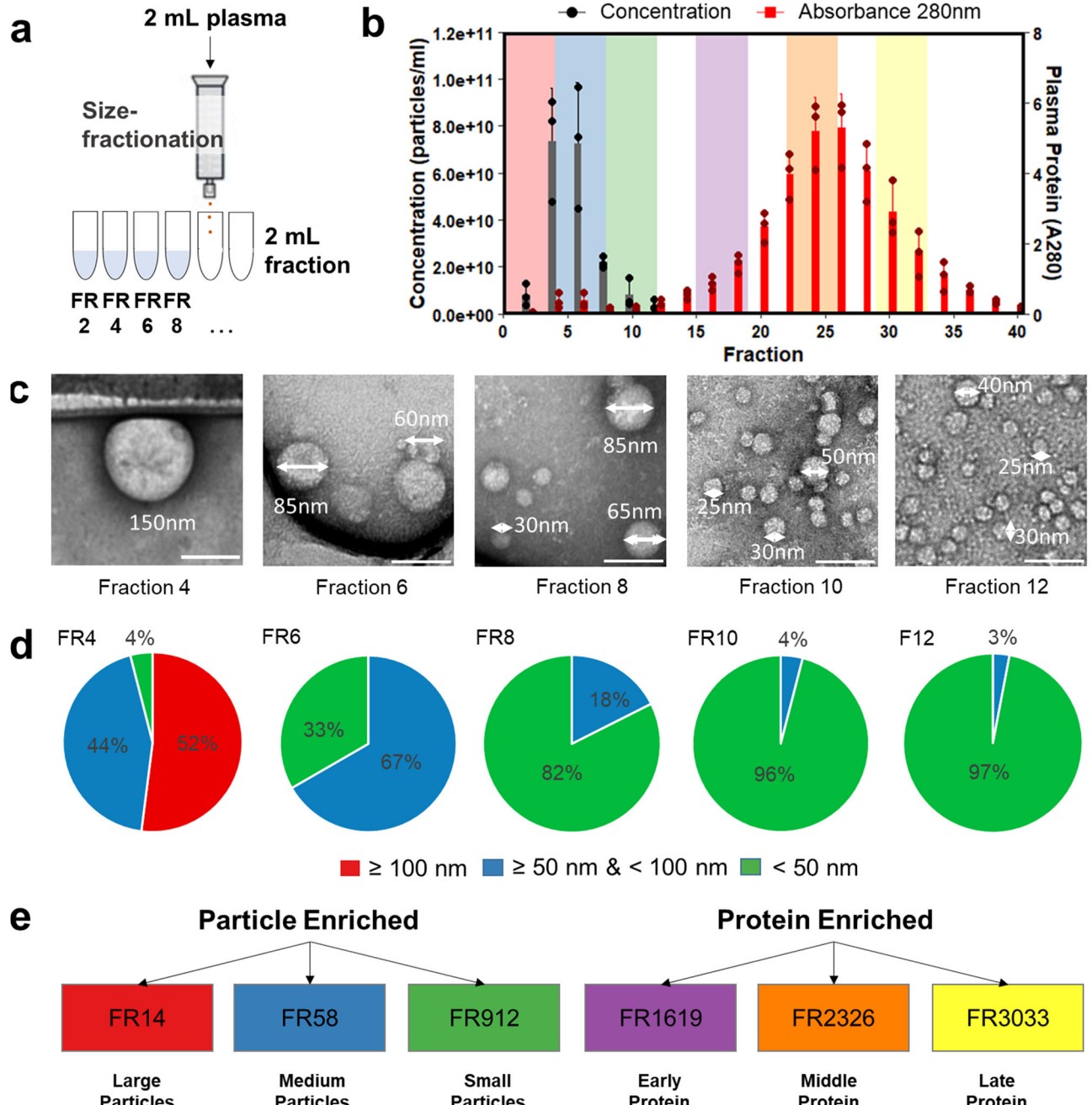

**Fig. 1 Characterization of human plasma by size fractionation. a** Schematic diagram of plasma fractionation using size exclusion chromatography with 2 mL input plasma and 2 mL eluted volumes collected per fraction (FR). **b** Bar graphs of particle concentration (in black) measured by tunable resistive pulse sensing on left y-axis, and protein abundance (in red) measured by absorbance at 280 nm on right y-axis. X-axis indicates each fraction from size exclusion chromatography with area of color shown for FR14 (red), FR58 (blue), FR912 (green), FR1619 (purple), FR2326 (orange), and FR3033 (yellow) respectively. Data are analyzed using RStudio (v2023.06.1 + 524) and reported as means ± SD of three biological replicates. **c** Representative transmission electron microscopy images of particles in individual fractions; scale bar: 100 nm. **d** Pie chart of percent distribution of particles with corresponding size ranges: ≥ 100 nm (red), ≥ 50 nm & < 100 nm (blue), and < 50 nm (green) identified in each fraction measured by TEM. **e** Schematic of plasma fractions with color schemes for transcriptomic analysis.

protein fractions (FR2326: eluted from 22 to 26 mL), and late-eluting protein fractions (FR3033: eluted from 29 to 33 mL) for further analyses (Fig. 1e).

**Transcriptomic analysis of fractionated plasma.** To characterize cf-mRNA associated with different fractions, we purified and sequenced RNA from the size-fractionated plasma of 5 healthy controls (HD), 5 lung cancer (LG), 5 multiple myeloma (MM),

and 5 liver cancer (LV) patients (Supplementary Data 2, Fig. 2a). We added the same amount of ERCC spiked-in mix to each fraction sample to control for technical variability and for normalization. RNA purified from plasma are short fragments with predominant peak of less than 200 nt shown by bioanalyzer analysis (Supplementary Fig. 3a). To confirm the distribution of plasma cell-free RNA length, we designed PCR primers targeting an RNA long amplicon of 898 bp and short amplicons of 80 bp located at 5' end, 3'end and the middle of the transcript of the

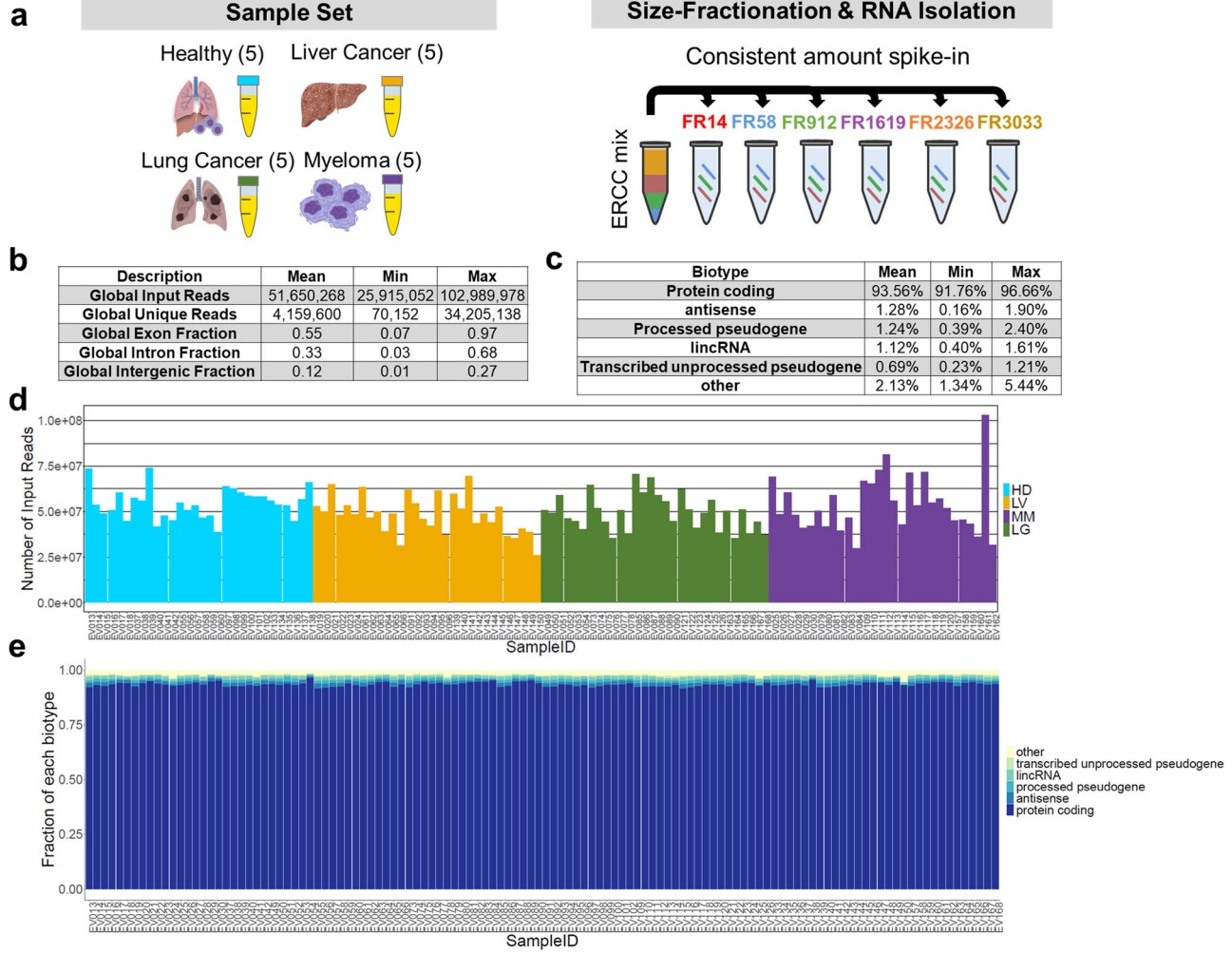

**Fig. 2 Transcriptomic analysis of size fractionated human plasma. a** Schematic of plasma fractionation workflow from the sample set. Plasma samples from 5 of each healthy control (HD) and cancer patients (LV, LG, MM) were collected for size-fractionation and RNA isolation. Each plasma sample was fractionated into large particles (FR14), medium particles (FR58), small particles (FR912), early-, middle- and late-eluting protein fractions (FR1619, FR2326, and FR3033 respectively). A consistent amount of ERCC RNA control mix was spiked into plasma fractions to control for processing and normalization. Representative images were generated using BioRender illustration Software and PowerPoint. **b** Table of description of global reads, unique reads, exon fraction, and intron fraction across 120 sequencing samples from RNA-seq quality control package (RSeQc). The mean, minimum (Min) and maximum (Max) values are shown. **c** Table of biotype categories including protein coding, transcribed unprocessed pseudogene, processed transcript, processed pseudogene, lincRNA, antisense, and others. The mean, minimum (Min) and maximum (Max) values are shown. **d** Bar graph of number of input reads across 120 sequencing samples colored by conditions. **e** A stack column representing the fraction of each biotype across 120 samples.

tissue specific gene ALB (Supplementary Fig. 3b). All short amplicons at three locations on the transcript were amplified while the long amplicons were not detected in cell-free RNA. Therefore, we used the Stranded SMART-Seq method instead of oligo-dT-based to prepare the libraries to efficiently capture fragmented cell-free RNA for sequencing. Using this method, we sequenced libraries to a depth of 25.9 M to 102.9 M reads (Fig. 2b, d) and obtained a mean of 4.0 M uniquely mapped reads in the range of 70,152–34.2 M for each sample (Fig. 2b). mRNA coding contigs were covered with sequencing reads (Supplementary Fig. 4). The percent of uniquely mapped reads declined as the fraction number increased: 17.8% (FR14), 9.9% (FR58), 4.2% (FR912), 5.2% (FR1619), 5.5% (FR2326), and 1.8% (FR3033) (Supplementary Fig. 5a). The fraction of reads mapping to exons (exon fraction) similarly decreased: 94.4% (FR14), 84.5% (FR58), 55.3% (FR912), 32.6% (FR1619), 25.5% (FR2326), and 36.3% (FR3033) (Supplementary Fig. 5b). While the percentage of uniquely mapped reads and exon fractions both declined as the fraction number increased, intron fractions increased in the same

samples (Supplementary Fig. 5c). Among the ~12,000 cfRNA transcripts found in fractionated plasma, cf-mRNA represented 93.6 ± 0.9% (mean ± SD) of the identified cell-free RNA biotypes, which remained consistent across different clinical sample types and fractions (Fig. 2c, e).

**Cf-mRNA is primarily enriched in large and medium particle fractions**. To gain insight into the relative abundance of cf-mRNA across plasma fractions, we filtered for protein-coding transcripts only and normalized by ERCC for further analysis (Fig. 3a). A total of 11,609 normalized protein-coding transcripts passed this threshold and were used for downstream analysis. Median log2 normalized expression of all detected protein-coding transcripts was 6 times higher in FR14 and FR58 compared with RNA contained in FR912 and protein-enriched fractions (Fig. 3b). Unsupervised principal component analysis (PCA) using the top 500 genes with the largest variance clearly separated large and medium particles from other plasma fractions and

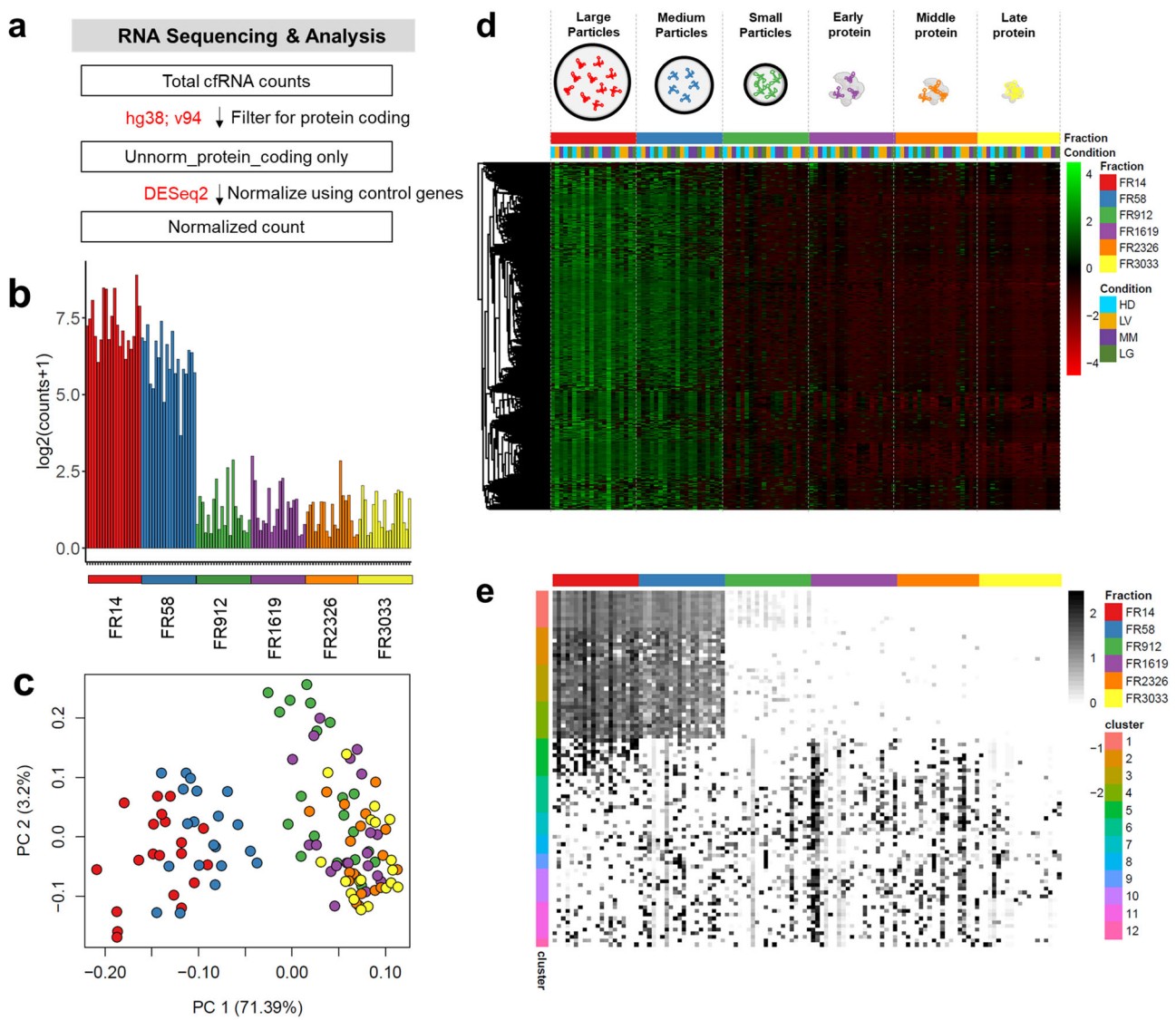

**Fig. 3 Majority of cf-mRNAs are present in large and medium particle fractions. a** Schematic of RNA sequencing and analysis workflow. Total cfRNA counts were filtered for protein coding using human genome ensemble annotation; v94 (hg38; v94), which were then normalized using ERCC as control genes by DESeq2. **b** Bar plot of median expression in log2(counts+1) for all detected genes across plasma fractions: FR14 (red), FR58 (blue), FR912 (green), FR1619 (purple), FR2326 (orange), and FR3033 (yellow) respectively. **c** Principal component analysis using all 11,609 detectable genes across individual fractions: FR14 (red), FR58 (blue), FR912 (green), FR1619 (purple), FR2326 (orange), and FR3033 (yellow) respectively. **d** Heatmap representing relative expression of all genes (n = 11,609) from all conditions across all 118 samples revealing that the majority of cell-free mRNA are enriched in FR14 and FR58. **e** Heatmap representing relative expression level of top 10 genes derived from 12 distinct clusters using degPatterns across 118 samples. Top 10 genes were ranked by False Discovery Rate (FDR) from one-way ANOVA test. Clusters with less than 10 genes were plotted with the actual number of genes.

showed a gradual transition from large and medium particles to small particles and protein fractions (Fig. 3c). Interestingly, the small particle fraction (FR912) exhibited a higher degree of similarity to protein-enriched fractions than large or medium particle-enriched fractions. A hierarchical clustering analysis using all detected genes (n = 11,609) revealed that the majority of cf-mRNA were enriched in large and medium particle fractions (Fig. 3d). This relatively high expression of cf-mRNA in large and medium particles is not a technical artefact since control synthetic ERCC RNA, which was spiked with the same amount across fractions, was consistent across all fractions after uniform sample processing (Supplementary Fig. 6).

To determine patterns of cf-mRNA detected in particle-enriched and protein-enriched fractions, we used the DegPatterns package to analyze 11,577 differentially expressed genes determined by one-way ANOVA test across all fractions. Using

DegPatterns, we identified 12 distinct trends whose inter-cluster variability is greater than intra-cluster differences (Supplementary Fig. 7). We selected the top 10 genes representative of each identified pattern for visualization in a heatmap. We observed four major enrichment patterns across plasma fractions: (1) 99.13% of genes (11,476 /11,577) were enriched in both FR14 and FR58 (cluster 1–4), (2) 0.38% of genes (44 /11,577) were enriched specifically in FR14 (cluster 5), (3) 0.04% of genes (5/11,577) were specifically enriched in FR58 (cluster 8), and (4) 0.45% of genes (52/11,577) were enriched in protein fractions (cluster 6-7, 9–12) (Fig. 3e and Supplementary Fig. 7).

**Cf-mRNA are enriched and protected in EVs**. We next examined molecular markers of potential RNA carriers such as EVs, lipoproteins and RNA-binding protein complexes across SEC

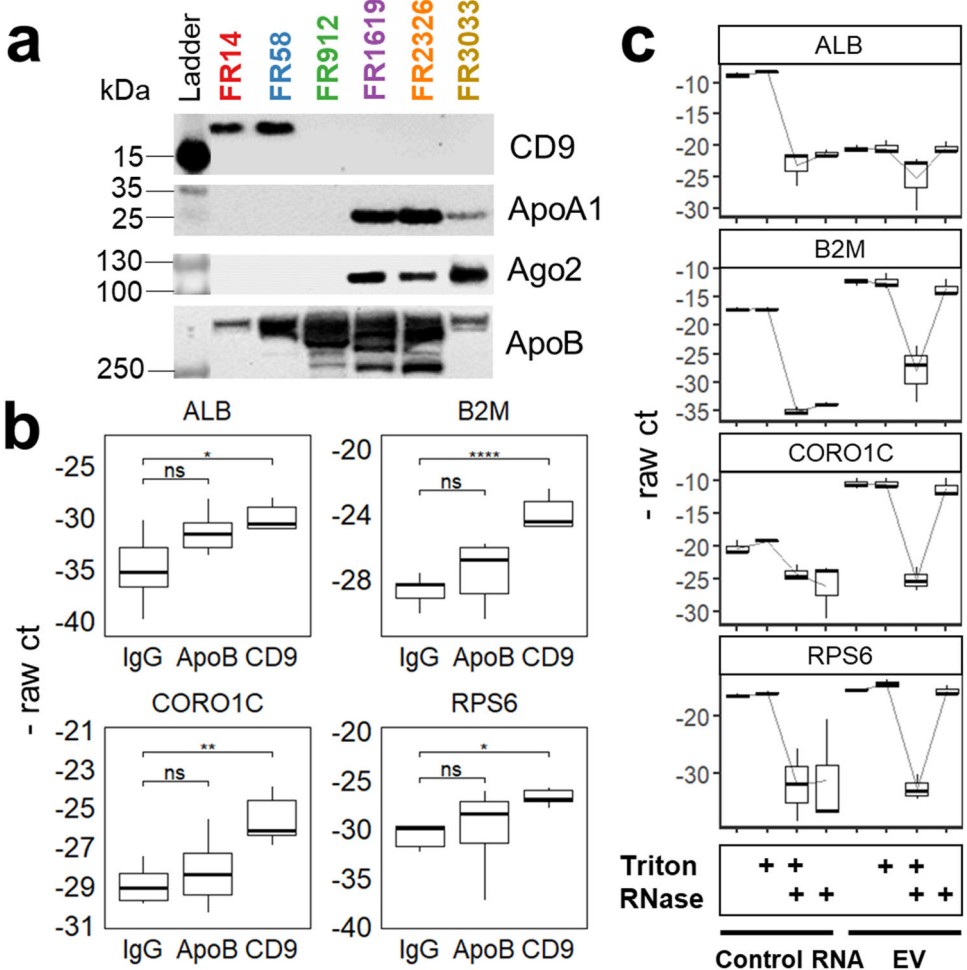

**Fig. 4 cf-mRNAs are enriched and protected in EVs. a** Expression of protein markers (CD9, ApoA1, Ago2, and ApoB) using immunoprecipitation across plasma fractions. Ladder was cropped from full gel to show size reference. **b** RT-qPCR analysis of selected genes using supernatants eluted from CD9, ApoB, or IgG immunocaptures from FR14 and FR58. *P*-value is derived from the Tukey's test (ns = not significant, *P* > 0.05; *$P$ < 0.05, **$P$ < 0.01, and ****$P$ < 0.0001) with three technical replicates of FR14 and FR58 of plasma pooled from three healthy individuals. **c** A box plot of negative raw cycle threshold (- raw ct) of individual genes with RNase and/or detergent treatment using qRT-PCR. RNA isolated from combined FR14 and FR58 using three healthy individuals and three technical replicates of control RNA were treated with RNase and/or detergent. Data are analyzed using RStudio and reported as means ± SD of three independent samples.

fractions. We performed immunoprecipitation on plasma fractions using antibodies against a canonical EV marker (CD9), lipoprotein markers (ApoA1 for HDL and ApoB for chylomicron, VLDL, and LDL), and one of the RNA-binding protein complexes, Argonaut 2 (Ago2) (Fig. 4a and Supplementary Fig. 8). Immunoprecipitation analysis showed that the canonical EV marker CD9 was preferentially enriched in FR14 and FR58. In contrast, ApoA1 and Ago2 were enriched in protein fractions (FR1619, FR2326, and FR3033). Furthermore, relatively large lipoprotein contribution was determined by using ApoB (Fig. 4a and Supplementary Fig. 8). Based on their physical size characteristics, EVs and similar size ranges of lipoprotein particles can be co-eluted[34]. Western blot analysis confirmed the highest levels of ApoB in the small particle fraction (FR912) and early-protein fraction (FR1619) while relatively low and moderate levels of ApoB were detected in FR14 and FR58 as well respectively. The small particle fraction (FR912) only showed ApoB expression, but CD9, ApoA1, and Ago2 were not detectable. To further examine the relative contribution of cf-mRNA from EVs and lipoproteins, we performed quantitative reverse transcription-polymerase chain reaction (qRT-PCR) analysis on cf-mRNA following immunoprecipitation with canonical EV marker CD9 and

lipoprotein marker ApoB from EV-associated fractions (FR14 and FR58). We chose 4 genes encompassing different cfRNA origins such as tissue specific gene (ALB), housekeeping genes (B2M), a back splicing junction (CORO1C), and ribosomal protein gene (RSP6). qRT-PCR analysis revealed that the majority of the tested genes were significantly enriched in CD9 immunoprecipitates compared to IgG control (Fig. 4b). However, no significant enrichment of tested RNAs in ApoB immunoprecipitates versus IgG control was found (Fig. 4b). Collectively, this data further confirmed that the major contribution of cf-mRNA in FR14 and FR58 is carried by EVs.

To further test if cf-mRNA is protected inside EVs from endogenous RNase in human plasma, we subjected FR14 and FR58 pooled fractions to RNase and detergent treatments followed by purification of RNA and quantification by qRT-PCR (Fig. 4c). We used RNA isolated from tissue (unprotected) as a control and the combined FR14 and FR58 with buffer alone, detergent Triton X-100 (expected to disrupt membrane-encapsulated EVs), RNase A alone, and both RNase A and Triton X-100. Total digestion of cf-mRNA from large and medium particle fractions only occurred following treating with both RNase A and detergent, consistent with cf-mRNA being

enclosed in vesicles (Fig. 4c). In contrast, relative gene expression for RNase A alone remained the same as detergent alone. Control RNA, which lacks any protective lipid bilayer, was totally digested by RNase A alone and RNase A with detergent (Fig. 4c). This observation supports the contention that cf-mRNA is primarily encapsulated within EVs. Taken together, our results indicate that cf-mRNAs are predominately present and protected in plasma within medium and small EVs. Considering these altogether: 1) maximum ApoB peaks were in FR912 and FR1619, while the majority of cf-mRNA was found in FR14 and FR58, 2) significant cf-mRNA was detected from CD9 immunocaptured EVs, but not from ApoB immunocaptured lipoproteins, and 3) RNase protection within lipid-bilayers, excluding possibility of cf-mRNA present on the surface, lead us to conclude that EVs are the major carrier of cf-mRNA in plasma. Therefore, hereinafter, we refer to FR14 and FR58 as medium EVs and small EVs, respectively.

**Selective enrichment of cancer differentiating cf-mRNA**. Next, we examined if mRNA contents in EV subpopulations are different between plasma from cancer patients and healthy controls. To identify RNA packaging associated with cancer, we identified differentially expressed (DE) genes (padj < 0.05 & log2FC > 1) between each cancer type and healthy controls within a specific fraction (Fig. 5a). We used permutation tests to evaluate the statistical significance of DE analysis (Supplementary Fig. 9). To analyze the enrichment patterns, we calculated fold change of these DE genes identified between each cancer type compared with controls within each fraction and visualized in a heatmap (Fig. 5a). We found that, compared to healthy controls, the fractionated plasma of lung cancer patients had significantly upregulated mRNAs, specifically 136, 199, 462, 151, 105, and 7 within medium EVs, small EVs, non-EV particles, early-, middle-, and late-eluting protein fractions respectively (Supplementary Fig. 10a). Intriguingly, the heatmap visualization of fold changes in cancer patients compared with healthy controls displayed distinct enrichment patterns of lung cancer-differentiating cf-mRNA, specific to EV and protein subpopulations (Fig. 5b). The majority of DE genes are enriched in specific EV and protein fractions with very few DE mRNA being shared between fractions (Supplementary Fig. 10). To assess the potential roles of these selectively enriched cancer distinguishing gene sets in each plasma fraction, we performed pathway enrichment analysis using g:Profiler curated on gene ontology and Reactome databases, Cytoscape, and EnrichmentMap (Fig. 5a). Differentiating genes enriched in FR14 (medium EVs) from lung cancer were associated with both immune system and metabolic process (Fig. 5c), while DE genes enriched in FR912 (non-EV particles fraction) were implicated in the immune system and myeloid leukocyte mediated response (Fig. 5c). Differentiating genes enriched in FR1619 (early-eluting protein) from lung cancer were enriched in protein localization nucleus, steroid hormone corticosteroid, and defense virus symbiont, while DE genes enriched in FR3033 (late-eluting protein) were enriched in sphingolipid metabolism (Fig. 5c).

We performed similar selective packaging and pathway enrichment analyses on plasma from patients with multiple myeloma and liver cancer. In multiple myeloma, we identified 37, 60, 102, 121, and 65 DE genes in medium EVs, small EVs, non-EV particles, early- and middle-eluting protein fractions, respectively (Supplementary Fig. 10b). We found no differentiating mRNA in late-eluting protein fractions. Multiple myeloma differentiating cf-mRNA revealed distinctive enrichment patterns specific to EV and protein subpopulations (Supplementary Fig. 11a). DE genes found in medium EVs (FR14) were related

to toxic substance detoxification (Supplementary Fig. 11b). FR2326 differentiating genes in multiple myeloma are enriched for protein regulation, neutrophil chemotaxis, and humoral immune response (Supplementary Fig. 11b). In liver cancer, we discovered 12, 109, and 38 DE genes in FR14, FR58 and FR912 fractions, respectively (Supplementary Fig. 10c). We found no differentiating mRNA in protein fractions. Similarly, our heatmap analysis revealed distinctive enrichment patterns of liver cancer differentiating cf-mRNA specific to medium EVs, small EVs and non-EV particles (Supplementary Fig. 12a). Liver cancer-distinguishing genes found in medium EVs (FR14) were related to the regulation of wound coagulation and negative regulation cell, while DE mRNA selective to small EVs (FR58) were related to regulation wound coagulation, remodeling protein complex, triglyceride metabolic processes, and plasma lipoprotein particles (Supplementary Fig. 12b). Taken together, our results identified a selective enrichment pattern of differentiating genes associated with different cancer types in EV subpopulations and protein fractions.

**Selective enrichment of multiple cancer differentiating cf-mRNA**. Having identified mRNA contents in EV subpopulations are different between plasma from cancer patients and healthy controls, we next sought to determine whether mRNA contents in EV and protein enriched fractions could distinguish between different cancer types. We analyzed the six plasma fractions (medium EVs, small EVs, non-EV particles, early-, middle-, and late-eluting protein fractions) of lung cancer, multiple myeloma, and liver cancer compared to those fractions in healthy controls. To identify RNA packaging associated with multiple cancers, we combined all of the differentially expressed (DE) genes (padj < 0.05 & log2FC > 1) per cancer type compared to healthy controls within each fraction. Intriguingly, our heatmap visualization of enrichment patterns revealed distinct multiple cancer differentiating cf-mRNA specific to EV and protein fractions (Fig. 6a). Although there is a slight increase in expression of cf-mRNA between small EV-associated lung cancer DE genes with multiple myeloma and liver cancer, the majority of these genes were not differentially expressed within multiple myeloma and liver cancer. In addition, we performed functional enrichment analyses on these combined gene sets and found the majority of biological function related terms were unique for each cancer type and fraction with minimal overlap on humoral response antimicrobial from both FR14 of lung cancer and FR2326 of multiple myeloma (Fig. 6b). These overall selective enrichment patterns associated with multiple cancers within EV subpopulations and protein fractions suggests that distinct cf-mRNA are packaged into different types of extracellular carriers in cancer.

## Discussion
Identifying cf-mRNA cargo types from human plasma and how different cancers dysregulate them is critical given emerging studies into their roles and functions. In this study, we investigated the association of cf-mRNA with potential biological carriers by fractionating plasma into large particles, medium particles, small particles, and protein-enriched fractions. To our knowledge, this is the first study to determine the association of plasma cf-mRNA with EV and non-EV carriers in human plasma. Among the cf-mRNAs detected and quantified by sequencing, 99% were present in large and medium particle fractions. Critically, we found that cf-mRNA from EV-associated fractions are also protected in lipid bilayers from RNase treatment. This finding suggests a mechanism by which cf-mRNAs can remain stable inside EVs from endogenous RNases in human plasma. Importantly, we identified not only that certain cf-

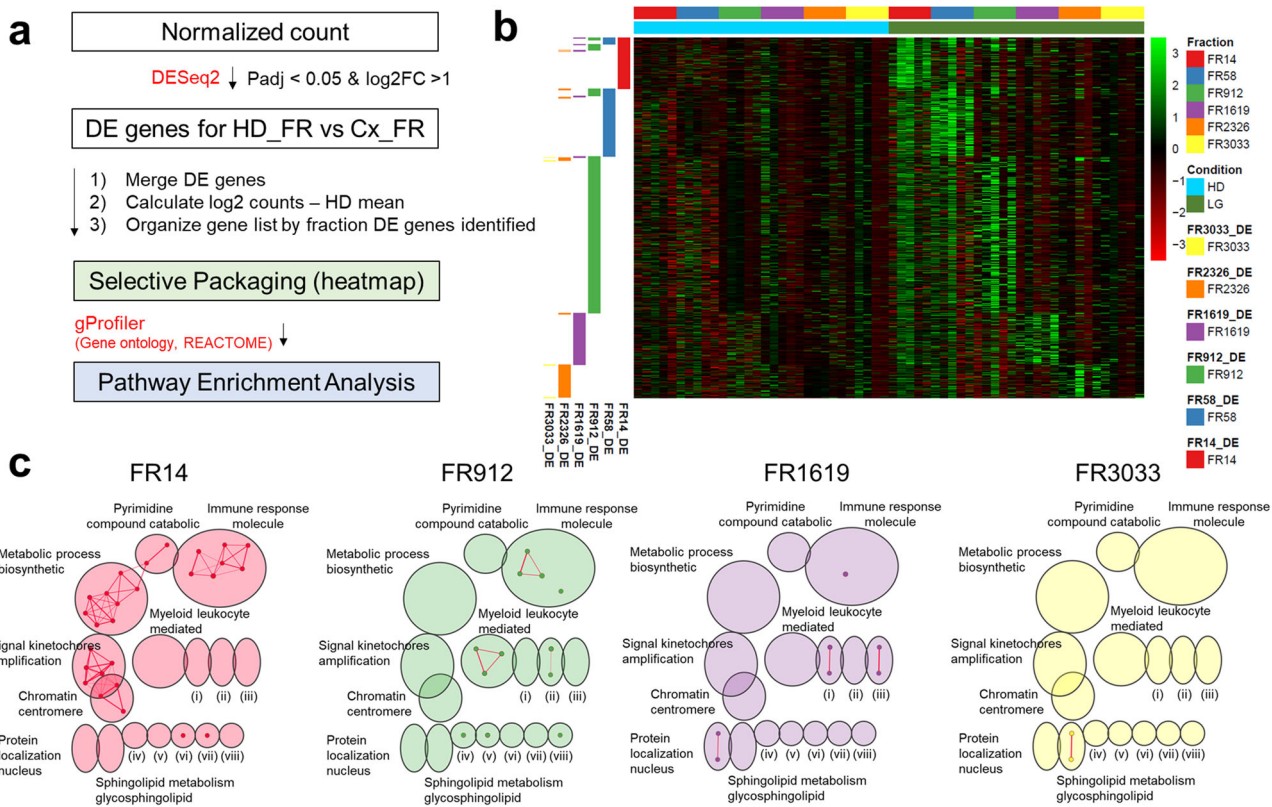

**Fig. 5 Selective Enrichment of Lung Cancer Differentiating cf-mRNA. a** Schematic of workflow for selective packaging (heatmap) and pathway enrichment analysis for lung cancer differentiating cf-mRNA. Normalized count was filtered to identify differentially expressed (DE) genes between individual healthy and cancer fraction (padj < 0.05 & log2FC > 1). After DE genes were identified, log2 fold change (log2FC) was calculated by subtracting log2 counts of cancer from individual biological replicates to the mean of corresponding healthy fractions. Gene lists were organized by fraction. DE genes were identified for selective packaging heatmap analysis. Using the DE genes identified from each fraction, g:Profiler was performed on gene ontology biological properties and reactome for functional enrichment analysis. Pathway enrichment analysis was performed using Cytoscape and EnrichmentMap. **b** Heatmap of gene expression in lung cancer relative to healthy across fractions. **c** Enrichment map for lung cancer DE genes found in individual fractions using Gene Ontology (Biological properties) and Reactome with FR14, FR912, FR1619, and FR3033 color coded by red, green, purple and yellow, respectively. Cluster represents (i) steroid hormone corticosteroid, (ii) Cellular response chemical, (iii) defense virus symbiont, (iv) regulation myeloid cell, (v) negative regulation response, (vi) toll cell receptor 4, (vii) g1 specific transcription, and (viii) hemidesmosome assembly. Cluster of nodes were automatically labeled using the AutoAnnotate from Cytoscape.

mRNAs are differentially expressed in cancer compared to healthy controls, but also that they contain functional sets of genes uniquely enriched in different cargoes.

To understand extracellular RNA communication in humans, it is important to identify the extent to which cf-mRNA are associated with vesicular and non-vesicular components. Murillo et al. utilized integrated computational analysis to find the major carriers of cf-mRNA within different biofluids[2]. Encompassing 23 healthy conditions across 19 studies, they found that cf-miRNA was associated with EVs, RNA binding proteins, or as part of lipoprotein particles, mostly HDL[2]. One of the major obstacles they encountered was large unexplained variability both within and across different cfRNA profiling studies. As a potential contributing factor, previous studies revealed how blood processing, EV isolation methods, RNA extraction methods, sequencing library preparations, and specific biofluid complexity can result in variations across studies[1,3,35,36]. Therefore, we profiled different cf-mRNA cargoes within the same human plasma samples using a uniform RNA extraction and sequencing library method. We utilized size exclusion chromatography (SEC), which was shown to co-isolate less soluble proteins with EVs compared with other methods[36]. By employing SEC followed by spike-in synthetic RNA during RNA extraction, we were able to reproducibly isolate defined RNA carriers and analyze their cf-mRNA contents.

While size-exclusion chromatography provides known reproducibility and is highly relevant to purifying EVs for liquid biopsy research, we observed overlapping particle size distributions between medium EV- and small EV-enriched fractions and a certain degree of lipoprotein carry-over. Therefore, we first denoted each fraction by the dominant particle size, as observed through direct TEM quantifications. Subsequent western blot analyses, however, revealed different distributions for: CD9+ EVs in FR14 and FR58, ApoB+ lipoproteins with the highest peak at FR912 and FR1619, and ApoA1 and Ago2 both found in later protein fractions (FR1619, FR2326, and FR3033). We also observed non-EV particles with size smaller than 50 nm being eluted in FR912 between EVs and protein fractions. This fraction did not display detectable levels of canonical EV markers, Ago2, nor ApoA1, yet did show high ApoB expression. James W. Clancy et al., defined additional nanoparticles smaller than 50 nm as potentially exomeres and supermeres[37]. While exomeres and supermeres have been shown to express Ago2[37,38], we did not detect Ago2 in FR912. Although the smallest particle sizes and early protein fractions are consistent with LDL, a more detailed molecular characterization of these particles is beyond the scope of this study and requires further investigation. An alternative fractionation strategy utilizing asymmetric-flow-field-flow-fractionation (AF4) has exhibited unique capabilities to separate EV

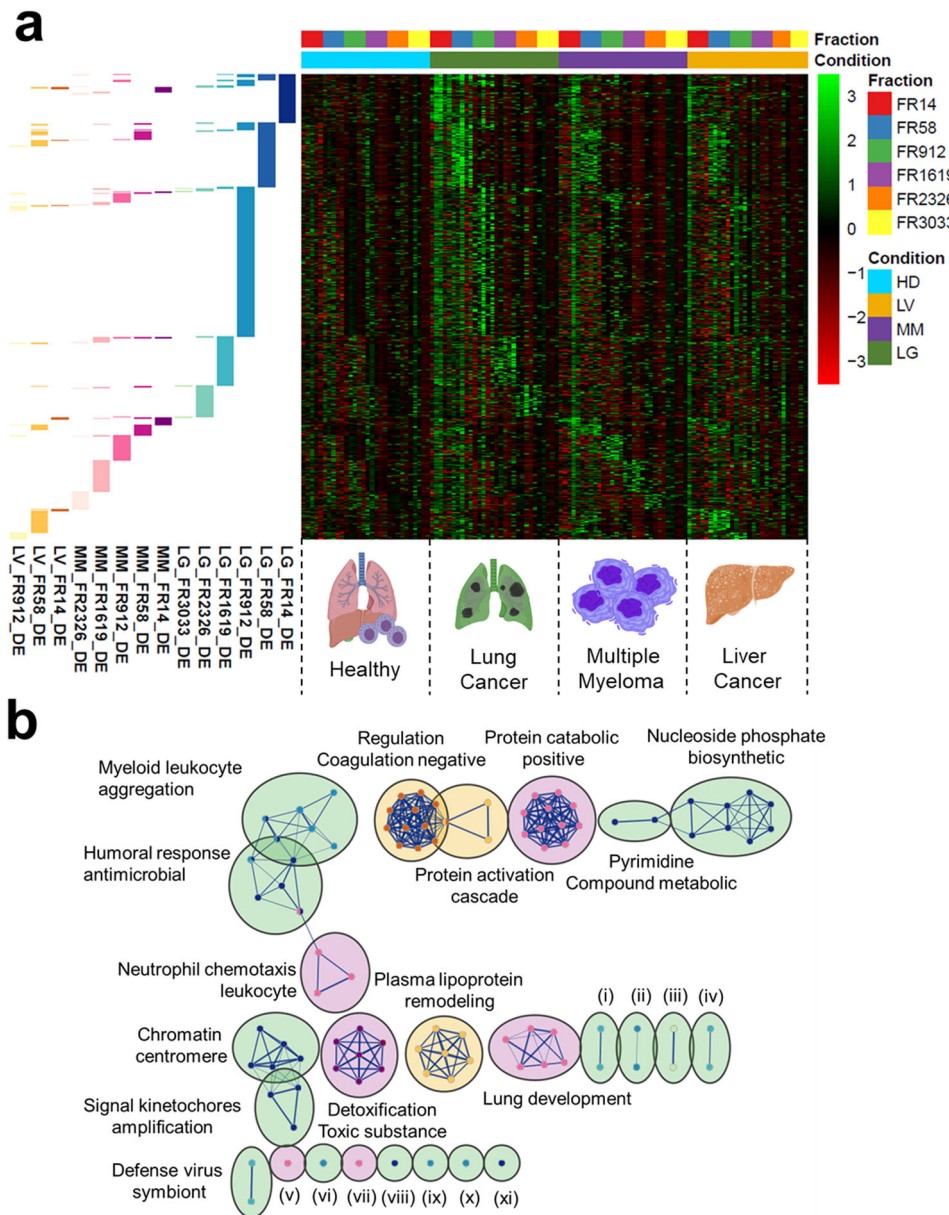

**Fig. 6 Selective Enrichment of Multiple Cancer Differentiating cf-mRNA. a** Heatmap of gene expression in multiple cancers (lung cancer, multiple myeloma, and liver cancer) relative to healthy across fractions. Annotation row indicates fraction of DE genes identified for specific cancer types compared to the healthy controls. Representative images were generated using BioRender illustration Software and PowerPoint. **b** Enrichment map for multiple cancer DE genes found in individual fractions using Gene Ontology (Biological Properties) and Reactome with lung cancer, multiple myeloma, and liver cancer color coded by green, purple and yellow, respectively. Cluster represents (i) steroid hormone corticosteroid, (ii) cellular response chemical, (iii) sphingolipid metabolism glycosphingolipid, (iv) localization nucleus nucleolus, (v) temperature homeostasis, (vi) regulation myeloid cell, (vii) alpha beta cell, (viii) toll cell receptor 4, (ix) hemidesmosome assembly, (x) negative regulation response, and (xi) g1 specific transcription. Cluster of nodes were automatically labeled using the AutoAnnotate from Cytoscape. Data sets were colored by fraction of DE genes identified for specific cancer types compared to the healthy controls as shown in Fig. 6a.

subpopulations from cell culture[39]. However, whether the AF4 method would reveal similar representative fractions in complex biofluids such as plasma remains unknown. More targeted studies demonstrated that miRNAs in normal human plasma could be immunoprecipitated by Ago2 antibodies[2,40], and yet methods like these provide an incomplete view of other carrier contributions.

A previous study indicated the presence of one micro RNA per 1–100 EVs and potentially even fewer for full-length mRNAs[41]. By cataloguing the whole cf-mRNA transcriptome from fractionated plasma samples, we captured the contributions from each carrier in plasma. However, without precise determination of the

purification and RNA extraction efficiency, the exact distribution of cf-mRNA copies per EV is not fully defined in this study. In addition, we have not accounted for possible variation in diet, exercise, circadian clock, gender, age, stage of diseases and donor-to-donor variations. However, in our previous study, we did not observe statistical difference in the total amount of cell-free RNA and the level of housekeeping transcript GAPDH for plasma cf-mRNA samples collected from the same individual throughout different times of day or between days[42]. Considering that meal consumption, which is an important factor that alters circulating lipoprotein levels, did not substantially change cf-mRNA levels

throughout the day, more mechanistic studies of how lipoproteins may carry cfRNA should be further explored. In addition, as lipoprotein amounts affect plasma particle concentration, fasting before blood processing should also be considered. In the context of cancer, more samples of patients along with careful consideration across categories of carriers may allow us to better understand the stage differences of the disease and also how different disease types may affect the concentrations of different sized EVs, lipoproteins, and associated cfRNA levels. Further studies will benefit from more precise size separation while providing reproducibility, accuracy and practicality to better investigate cf-mRNA's biomarker potential and even mechanisms behind cf-mRNA sorting into EVs in plasma.

Here, we demonstrated that the majority of cf-mRNA in plasma (~99%) were associated with large and medium particle fractions. While we observed a significant difference in relative log expression of mRNA between particle and soluble plasma protein enriched fractions, samples from fractionated plasma samples were sequenced to a uniform depth. Importantly, synthetic RNA spike-in controls enabled a standardized assessment of the relative cf-mRNA content across fractions, which would otherwise have been falsely amplified or regressed by a traditional library-size normalization approach[43,44]. A similar study using size-fractionation, yet which focused on miRNA, by Arroyo et al. found that Argonaut2 was coeluted with miR-16, miR-92, and miR-122 in protein enriched fractions[12]. Notably, they found that some miRNAs that might originate from cell types known to generate vesicles, like let-7a, were preferentially detected in EVs[12]. Despite general similarities in methods used for characterizing cf-miRNA or cf-mRNA carriers, different RNA biotypes may be associated with different carriers in plasma.

Narrowing down the potential carrier of observed cf-mRNA enriched in large and medium particles was a key aspect of our study. RNase treatment analysis with and without membrane disruption by detergent suggested that the majority of cf-mRNA in plasma are protected from degradation inside of EVs. Our findings are consistent with Enderle et al., who found that the relative quantification of column-bound cf-mRNA decreased after combined treatment with RNase and detergent[45]. Inclusion of a non-membrane bound form of cf-mRNA included in our study was an important control as it was readily reduced with RNase both with and without detergent treatment. These findings support that the majority of cf-mRNAs are protected inside of lipid bilayers from RNase-rich environments like plasma. In addition, we examined the abundance of lipoproteins (ApoA1 and ApoB), and RNA-binding proteins (Ago2) using immunoprecipitation. While ApoA1 and Ago2 were eluted separately in protein-enriched fraction, ApoB was partially cofractionated within EV-associated fractions. By performing immunoprecipitation followed by qRT-PCR, we further proved that cf-mRNA is significantly enriched in CD9 immunoprecipitates compared with control IgG, while none were significantly enriched in ApoB immunoprecipitates compared with control IgG. Understanding the mechanisms of specific gene packaging differences and its association with cancer to utilize as liquid biomarkers remain as the subject of our future studies.

We observed that the cf-mRNA contents between various cancers and healthy controls were altered between different RNA carriers. Previous reports have shown functional delivery of cfRNA by EVs can promote tumorigenesis, invasion, and cell proliferation[46,47]. In particular, cf-mRNA content of EVs showed a remarkable difference in clinical human samples. Ramshani et al. developed a surface acoustic wave (SAW) EV lysing microfluidic chip and found that the concentration of miR-12 was nearly 13-fold higher in liver cancer patients[30]. Interestingly, the presence of mutations on oncogenes such as KRAS has been shown to suppress Ago2 interactions with endosomes, resulting in different miRNA incorporation into EVs[28]. In our study, we revealed distinct enrichment patterns in both particle and protein enriched fractions associated with cancer from human plasma. Interestingly, functional characterization of these distinct differential gene sets across specific plasma fractions contained both shared and unique biological pathways. In particular, lung cancer-distinguishing genes identified in medium EVs, non-EV particles, and early-eluting protein fractions were associated with immune response molecule. Liver cancer-distinguishing genes in small EVs were associated with plasma lipoprotein particles, which are related to liver function[48]. Furthermore, multiple myeloma-distinguishing genes in medium EVs were associated with toxic substance detoxification, potentially related to the dysregulation of erythropoiesis in multiple myeloma[49]. Given the ability to detect unique cf-mRNA in different carriers, future mechanistic studies are needed to examine how specific oncogenic drivers could impact selective cf-mRNA packaging due to cancer. In addition, further studies are needed to examine the mechanisms governing cf-mRNA sorting into plasma EVs and other carriers, their cell types and tissues of origin, and whether cf-mRNA sorted into EV subpopulations in the blood have signaling functions.

## Methods

**Clinical samples and plasma preparation.** Blood samples from control individuals and patients with multiple myeloma, liver cancer, and lung cancer were obtained from Oregon Health and Science University (OHSU) by Knight Cancer Institute Biolibrary and Oregon Clinical and Translational Research Institute (OCTRI). All samples were collected under OHSU institutional review board (IRB) approved protocols. All donors gave written informed consents for research use, and all relevant ethical regulations were followed. Control donors were individuals with no known previous history of cancer. All samples were collected and processed using a uniform protocol by the same staff at OHSU. Samples for analysis were matched between cancer and control groups with respect to age and gender of participants. We previously comprehensively tested the pre-analytical variation of cf-mRNA depending on blood collection tubes and processing protocol[35]. Based on that result, we established the following blood processing methodology to limit variability in ex-vivo generation of EV subpopulations and cf-mRNA transcripts associated with platelet activation. Whole blood was collected from all clinical samples in 10 mL K2EDTA tubes (BD Vacutainer, Becton Dickinson, 36643). Tubes were transported vertically at room temperature before processing. Within 1 h of blood withdrawal, plasma was prepared by differential centrifugation (Eppendorf 5810-R centrifuge, S-4-104 Rotor, Eppendorf). First, 10 mL of whole blood was spun at $1000 \times g$ for 10 min at 4 °C. The supernatant was collected 10 mm above the buffy coat. A second centrifugation was performed at 15,000 x g for 10 min at 4 °C. Aliquots of platelet-depleted plasma were transferred to 1.5 mL microcentrifuge tubes (VWR, 89126-714) and stored immediately at −80 °C.

**Plasma fractionation using size exclusion chromatography.** Size exclusion chromatography was conducted using commercially available qEV2 SEC columns (Izon Science Ltd, New Zealand, SP8-USD) according to the manufacturer's instructions. In brief, the column was equilibrated with 0.1 μm filtered D-PBS without calcium and magnesium (Gibco, 14190250) at room temperature. Two mL plasma was loaded onto the column, and 14 mL of void volume was discarded. Exactly 4 mL of D-PBS were added into SEC columns using a 5 mL pipette to collect each

plasma fraction. We collected 6 plasma fractions per each sample from SEC columns on ice in separate 50 mL canonical tubes (Corning, 14-432-22): large particles (FR14: eluted from 0 to 4 mL), medium particles (FR58: eluted from 4 to 8 mL), small particles (FR912: eluted from 8 to 12 mL), early-eluting protein fractions (FR1619: eluted from 15 to 19 mL), the middle-eluting protein fractions (FR2326: eluted from 22 to 26 mL), and late-eluting protein fractions (FR3033: eluted from 29 to 33 mL). The collection of these six plasma fractions was followed immediately by RNA extraction.

**EV concentration and plasma protein measurements**. Concentrations of particles in isolated particle-enriched fractions obtained from SEC using plasma from three healthy individuals were measured using tunable resistive pulse sensing by qNano (Izon, Cambridge, MA, USA) according to the manufacturer's instructions. Due to the limited availability and amount of blood samples for cancer patients, we only performed in healthy individual's samples. First, nanopore (Izon, NP150) was placed on a fluid cell and stretched to 47.0 mm to calibrate the stretch. After adding wetting solution to the lower and upper fluid cell with pressure of 20 mbar for 4 min, we established a stable baseline current at approximately 120 nA. Wetting solution was replaced by coating solution, and maximum pressure and vacuum was applied at 20 mbar for 10 min each. Coating solution was flushed out of upper and lower fluid wells three times with measurement electrolyte, and maximum pressure was applied for 10 min to equilibrate the measurement electrolyte. After removing the electrolyte, we added 110 nm calibration particles at $1 \times 10^{13}$ particles per mL (Izon, CPC100) to optimize stretch, voltage, and pressure so that relative particle size is within 0.25–0.5% with its speed within 10–15 /ms. After we recorded calibration beads, we placed the particle-enriched fractions collected from SEC on the nanopore (Izon, NP150) and recorded them at the identical system settings as the calibration beads. The particle concentration of each plasma fraction was determined by converting the blockade frequency of each plasma fraction using Izon Control Suite (v3.3.2.2001, Izon Science). Plasma protein absorbance was measured at 280 nm using NanoDrop™ One/OneC Microvolume UV-Vis Spectrophotometer (Thermo Scientific, ND-ONE-W).

**EV size distribution measurement using TEM**. Ultrathin carbon film on lacey carbon support with 400 mesh on copper (Ted Pella, 01824) was glow discharged for 30 s using PELCO easiGlow glow discharger (Ted Pella). Isolated particle-enriched fractions from healthy individuals were put on charged grids for 1 min, washed for 30 s with MilliQ water, and fixed with 1% uranyl acetate for 30 s. Grids with stained samples were air dried for at least 30 min before imaging. Prepared samples were imaged at 120 kV using FEI Tecnai™ Spirit TEM system. FEI- Tecnai™ Spirit TEM system was interfaced to a bottom mounted Eagle™ 2 K TEM CCD multiscan camera and to a NanoSprint12S-B cMOS camera from Advanced Microscopy Techniques (AMT) fast side mounted TEM CCD Camera. Images were collected at 30,000x magnification under 3 μm defocus. Images were acquired as 4000 × 3000 pixel, 16-bit gray scale files using the Nanosprint 12 AMT interface and camera. For the analysis, TEM images were acquired with at least 25 particles were measured from each plasma fraction. Observed particles per fraction were binned with 5 nm width for the histogram. Particle diameters were manually measured using Fiji line-tool, and exported to excel/R for downstream analyses.

**Immunoprecipitation and western blotting**. For immunoprecipitation, 200 μL of Magna Bind goat anti-mouse IgG Magnetic

Bead slurry (Thermo Scientific, PI21354) was washed with PBS solution and incubated with 5 μg of mouse monoclonal anti-Ago2 (Abcam, ab57113), anti-CD9 antibody (Abcam, ab58989), anti-Apolipoprotein A1 (Santa Cruz Biotechnology, sc-376818), ApoB antibody from Santa Cruz (sc-13538) or mouse normal IgG antibodies (Santa Cruz Biotechnology, sc-2025) for 2 h at 4 °C. To account for smaller volumes for immunoprecipitation procedures, the 4 mL of 6 fractions (FR14, FR58, FR912, FR1619, FR2326, and FR3033) of SEC column were concentrated by ultracentrifugation at 150,000 g x 6 h. The resulting pellets were each lysed in 200 μL of IP lysis buffer (Thermofisher Scientific, 87787) supplemented with a halt protease inhibitor cocktail (Thermofisher, 78430). A total of 200 μL of IP lysed samples were mixed with 200 μL of PBS. The preincubated beads and antibody were then added to the 400 μL of sample and incubated overnight at 4 °C. Beads were washed three times with 1% Nonidet P-40 buffer (Sigma Aldrich, 11332473001) and then eluted in 20 μL of NuPage LDS/reducing agent mix and incubated for 10 min at 70 °C to elute the sample. Samples eluted off from the beads were used for western blotting. Western blots were run using Bolt 4–12% Bis-Tris Plus gel (Life technologies, NW04122) and transferred onto PVDF membrane (Thermofisher scientific, LC2002). The membrane was blocked with 1X TBST containing 5% milk (Sigma Aldrich, M7409-5BTL) and incubated with primary antibodies overnight at 4 °C. The anti-Argonaut-2 antibody (Abcam, ab32381), anti-CD9 (Abcam, ab223052), anti-Apolipoprotein A1 (Abcam, ab64308) and goat polyclonal anti-ApoB100 antibody (R&D Systems, AF3260) were used as primary antibodies. After washing with 1X TBST, the membrane was incubated with horseradish peroxidase conjugated anti-rabbit secondary antibodies (Cell Signaling, 7074) or anti-goat secondary antibodies (Promega, V8051) respectively, and washed again to remove unbound antibody. Bound antibodies were detected with Supersignal West Pico Plus Chemiluminescent Substrate (Thermofisher, 34577).

**RNase digestion of EVs and control RNA**. EVs encompassing large particles (FR14: eluted from 0–4 mL) and medium particles (FR58: eluted from 4–8 mL) were isolated from SEC column using 2 mL input plasma. The EV isolation step was repeated to collect a total of 16 mL of fractions with EVs, which were equally divided into four samples. As a control, 4 mL of lung tissue RNA (Takara Bio, 636524) at a final concentration of 500 pg/μL was used. Subsequently, 4 mL of either isolated EV or control RNA sample was mixed with 0.2% of triton X-100 (Sigma Aldrich, T8787) and/or 25 μg/mL RNase (Thermo Scientific, EN0531). After 30 min of incubation at room temperature with occasional swirls, RNA from EV or control RNA sample was extracted.

**RNA extraction from fractionated plasma**. RNA was extracted from 4 mL of fractionated plasma using a plasma/serum circulating and exosomal RNA purification kit (Norgen Biotek, 42800) according to the manufacturer's protocol with the following modifications. After fractionated plasma samples were lysed at 60 °C for 10 min and mixed with ethanol in step 2 of the procedure, 10 μL of $10^6$ times diluted ERCC RNA spike-in control mix (Thermofisher, 4456740) was added into each denatured plasma fraction sample (i.e. after combining the plasma fraction samples with denaturing solution in step 1) on ice as an external RNA control for normalization. These ERCC RNA-spiked in plasma fraction samples were followed by centrifugation at 1000 RPM for 2 min. After that point, we followed the manufacturer's protocol and eluted RNA in 100 μL. To digest trace amounts of contaminating DNA, we treated the RNA with 10X Baseline-ZERO DNase (Lucigen, DB0715K). DNase I treated RNA samples

were purified and further concentrated using RNA clean and concentrator-5 (Zymo Research, R1014) according to the manufacture's protocols. Final eluted RNA was aliquoted and stored at −80 °C immediately.

**Bioanalyzer and RT-PCR gel of cf-mRNA fragment analysis**. Cell-free RNA extracted from 2 mL of healthy individual plasma and 1 ng/μL of control universal human normal tissue RNA (Qiagen, C51105) were analyzed using Agilent 2100 Bioanalyzer with RNA 6000 Pico series kit (Agilent, 5067-1513). For cf-mRNA fragment analysis, PCR primers were designed to amplify gene fragments of different lengths across the gene ALB. One primer pair was designed to amplify a long 898 bp fragment of the gene, and 3 primer pairs were designed to amplify short gene fragments in the 5' end, middle, and 3' end of the gene. Expected amplicon sizes for these primers were 80 bp, 82 bp, and 76 bp respectively. Primer-sequences are as follows: long forward: AGAGTGAGGTTGCTCATCGG; long reverse: GGCAAGTCAG CAGGCATCTC; short 5' end forward: TCCCTTCTTTTTCTCT TTAGCTCG; short 5' end reverse: CGATGAGCAACCTCACTC TTG; short middle forward: GCTGAGGCAAAGGATGTCTTC; short middle reverse: GCAGCAGCACGACAGAGTAA; short 3' end forward: GCAAGGCTGACGATAAGGAGA; short 3' end reverse: CCTAAGGCAGCTTGACTTGCAG. Reverse transcription was performed separately for each primer pair using Invitrogen SuperScript III One-Step RT-PCR kit (Invitrogen, 12574026) and a final primer concentration of 0.1 μM, with 18 cycles of pre-amplification. After reverse transcription, samples were diluted 1:10 in ultrapure water. 1 μL of 1:10 diluted RT-preamplification product was used as the input for PCR, which was performed using 2X PCR master mix (Thermo Scientific, K0171) and a final primer concentration of 0.1 μM. 30 cycles of amplification were performed followed by a 10-minute final extension step. Following PCR, products lengths were assessed via gel electrophoresis, using a 2% agarose-TBE gel run at 70 V for 1 h. Low molecular weight DNA ladder (NEB, N3233L) was included as a size reference.

**qRT-PCR profiling of cell free mRNA from EV enriched fractions**. For immunoprecipitation followed by qRT-PCR, ultracentrifugation pellets from FR14 and FR58 were resuspended into 600 μL of D-PBS. The 600 μL were split into 3 subsets of 200 μL, which was incubated with anti-CD9, anti-ApoB, or anti-IgG for overnight with rotation at 4 °C. Next, beads were incubated in 30 μL 0.2 M glycine, pH 2.5 (Thermo Scientific, AAJ61855AK) for 15 min and magnetized for elution. Eluted fractions were neutralized with 5 μL of Tris-HCl, pH 8.6. Finally, 2 μL of the eluted product was used as the starting material for qRT-PCR. We selected 4 primers for genes which encompass a wide variety of cf-mRNAs such as tissue specific genes (i.e. ALB), housekeeping genes (B2M), a back splicing junction (CORO1C), and ribosomal protein gene (RSP6) that were optimized in-house. The 2 μL of the eluted product was mixed with Superscript III One-step RT-PCR system with Platinum Taq DNA polymerase (Invitrogen, cat. 11-732-020) to generate cDNA according to the protocol. cDNA from preamplification was diluted 1:80 and set-up in 96-well plates with SsoFast EvaGreen supermix with low ROX (BioRad, cat. 1725211) with above primers at 10 μM. QuantStudio 7 Flex (Applied Biosystems) was used to run RT-qPCR assay according to manufacturer's recommended cycling conditions with 40 cycles of amplification as optimized in our previous study[35].

**Sequencing library preparation**. We prepared stranded RNA-Seq libraries for each plasma fraction using Clontech SMARTer

stranded total RNA-seq kit v2- pico input mammalian (Takara Bio, 634414) according to the manufacturer's instructions. For cDNA synthesis, we used option 2 (without fragmentation), starting from highly degraded RNA. Input of 7 μL of RNA samples were used to generate cDNA libraries suitable for next-generation sequencing. For addition of adapters and indexes, we employed SMARTer RNA unique dual index kit −96 U (Takara Bio, 634452). SMARTer RNA unique dual index of each 5' and 3' PCR primer were added to each sample to distinguish pooled libraries from each other. The amplified RNA-seq library was purified by immobilization onto AMPure XP PCR purification system (Beckman Coulter, A63881). The library fragments originated from rRNA and mitochondrial rRNA were treated with ZapR v2 and R-Probes according to manufacturer's protocol. For final RNA-seq library amplification, 16 cycles of PCR were performed and the final 20 μL was eluted in Tris buffer following amplified RNA-seq library purification. The amplified RNA-seq library was stored at −20 °C for sequencing.

**RNA sequencing and data processing**. All fractionated plasma samples isolated by SEC were randomized to reduce sample batch effects and were uniformly processed for RNA extraction, library preparation, and sequencing in Illumina flow cells. All 120 pre-made RNA-seq library samples were sequenced using the NovaSeq 6000 system (Novogene, Sacramento, CA). The pre-made RNA-seq library samples were equally distributed over three NovaSeq S4 lanes for paired-end read x 150 bp sequencing. The quality of the reads were checked using FastQC (v0.11.8)[50,51] and RSeQC (v3.0.0)[52]. Reads were aligned to the human genome assembly (hg38, ensembl annotation; v94) and ERCC RNA spike-in sequences using the STAR aligner (v2.5.3a)[53] with two pass mode flag. Following adapter trimming and alignment using STAR (ver 2.5.3), bigwig coverage tracks were generated from each sample alignment file using bedtools (ver 2.27.1). Sample bigwig files were then visualized using the ggcoverage R package (ver 0.7.1) over the genomic range of the housekeeping gene ACTB as well as ALB. Read counts for each gene were calculated using the htseq-count tool (v0.11.2)[54] in intersection-strict mode. For each sample, we calculated exon, intron, and protein coding fractions (CDS exons) using RSeQC (v3.0.0)[52].

**RNAseq data analysis**. Among a total of 6 fractions (large particles, medium particles, small particles, early-, middle-, and late-eluting protein fractions) with 20 samples (including cancer and control biological replicates), we retained total cell-free RNA counts with more than five reads in at least one plasma fraction from all samples. This resulted in a total of 12,671 (out of 58,768) total cell-free RNA features and 55 (out of 92) ERCC spike-ins passing this threshold. The total cell-free RNA counts were then filtered by protein-coding biotype using human genome assembly (hg38, ensembl annotation; v94), resulting in identification of 11,609 expressed cell-free mRNA transcripts. The unnormalized protein coding transcripts were then normalized using ERCC RNA spike-in control as size factors in DESeq2 (v1.22.2)[55]. The relative log expression was obtained from RUVseq package (v1.16.1)[44]. 2 out of 120 of the plasma fractions were excluded due to the being an outlier through clustering of genes by expression pattern. Differential expression analysis between case/control of fractionated samples were analyzed using DESeq2 with adjusted p-value (padj) < 0.05 and log2 fold change (log2FC) > 1. To test the significance of the differential expression results for each pairwise comparison of cancer to healthy control per fraction, we performed a permutation test in which differential expression analysis between two groups of randomized samples was compared using the DESeq2 package. For each pair, 1000

permutations of random shuffling were performed and the number of differentiating genes with padj < 0.05 were documented for building a histogram, and compared to the number of significant genes (padj < 0.05) for the group with correct labeling.

**Pathway enrichment analysis**. Pathway enrichment analysis was performed according to Reimand et al. [56]. We created generic enrichment map (GEM) files for pathway analysis using gprofiler, which performs functional profiling of gene list from large-scale experiment (https://biit.cs.ut.ee/gprofiler/)[57]. An input (un-ranked) gene list was derived from differential expression analysis performed between case/control of fractionated samples analyzed through DESeq2. In addition to Gene Ontology (molecular function, cellular component, or/and biological process), we include pathways from Reactome. For visualization and network enrichment analysis, GEM files were imported to Cytoscape (v3.9.0) (http://www.cytoscape.org/), and GMT file from the g:Profiler website containing data source was specified. EnrichmentMap in Cytoscape (http://www.baderlab.org/Software/EnrichmentMap) was used to build the network[56], where nodes (circles) define the pathway. For number of nodes, genes were filtered by FDR q-value cutoff at 0.01. Individual biological themes were defined using AutoAnnotate in Cytoscape application.

**Statistics and reproducibility**. Data are reported as means ± SD of at least three independent experiments. Differentially expressed genes between case/control of fractionated samples were identified using adjusted p-value (padj) < 0.05 and log2 fold change (log2FC) > 1 from DESeq2 (v1.22.2). For DegPattern analysis from R package DEGreport (v1.18.1), a total of 11,577 differentially expressed genes determined by one-way ANOVA test across plasma fractions with false discovery rate less than 0.05 were used.

**Reporting summary**. Further information on research design is available in the Nature Portfolio Reporting Summary linked to this article.

## Data availability

Sequencing data (GSE205301) is deposited in the Gene Expression Omnibus Repository[58]. All numerical source data used in this manuscript are publicly available on Zenodo repository: https://doi.org/10.5281/zenodo.8211794[59]. All other data are available from the corresponding author on reasonable request.

## Code availability

In-house scripts used in this manuscript for graphs and analyses, which include data processing, downstream analysis, and the scripts used to generate figures are publicly available on Zenodo repository: https://doi.org/10.5281/zenodo.8211794[59].

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

## Acknowledgements

Research in Ngo lab was supported by the Cancer Early Detection Advanced Research (CEDAR) Center at OHSU Knight Cancer Institute (Full 2020-1289), Cancer Research UK/OHSU Project Award (C63763/A27122 to T.T.M.N.), the Kuni Foundation, the Department of Defense (W81XWH2110853 to T.T.M.N.) and the Susan G. Komen Foundation (CCR21663959 to T.T.M.N.). Sample collection was partly supported by the OCTRI CTSA grant (UL1TR000128). The authors acknowledge the assistance of multiscale microscopy core for transmission electron microscopy infrastructure and thank Claudia Lopez for providing assistance in microscopy. The authors thank Drs. Lorena Pantano, Theresa Lusardi for their helpful discussions. The authors thanks Conner Bailey for assisting RT-qPCR experiments. We would like to acknowledge the CEDAR repository and the Biolibrary for helping with sample collection and processing. H.K, M.J.R, F.C., C.W.K., B.R.H, E.S., J.B., A.D., J.E., E.D., and T.T.M.N. are members of and supported by the Cancer Early Detection Advanced Research (CEDAR) Center of the OHSU Knight Cancer Institute. Parts of Figs. 2, 6 were created with BioRender.com.

## Author contributions

H.K. and T.T.M.N. initiated the project. H.K. and M.J.R. performed wet-lab experiments including plasma fractionation and RNA extraction. H.K., M.J.R., F.G., and J.B. performed immunoprecipitations and western blotting. H.K., M.J.R., C.W.K., and E.S. performed qRT-PCR and analysis. H.K. and M.J.R. performed TEM experiments and visualization. H.K. performed sequencing library preparations and data visualization. H.K. and T.T.M.N. designed the experimental methodology. H.K., B.R.H., A.D., J.E., E.D., and T.T.M.N. designed the sequencing methodology and analysis. T.T.M.N. supervised the project. H.K. and T.T.M.N. drafted the initial manuscript, which was revised by H.K., T.T.M.N., G.M., M.J.R., J.T.W., B.R.H., J.B., A.D. and J.E.

## Competing interests

The authors declare no competing interests.
