## [Peer Review File · Communications Biology]

Reviewers' comments:

Reviewer #1 (Remarks to the Author):

The manuscript by T.T.M. Ngo and colleagues, "Selective enrichment of plasma cell-free mRNA in cancer-associated extracellular vesicles," presents a highly detailed and informative view of cell free mRNA in plasma. Moreover, the authors identify specific cf-mRNA signatures associated with multiple cancers, including lung cancer, liver cancer, and multiple myeloma. The results of this manuscript beautifully demonstrate how "dissecting," or fractionating, the extracellular carriers of mRNA facilitates the identification of certain disease-specific cfRNA signatures. The authors' approach is distinctive both in its fractionated analysis of extracellular RNA carriers and in the use of a stranded SmartSeq NGS approach to identify cell-free mRNA, along with the use of ERCC spike in controls for exRNA normalization across highly variable samples.

1. The authors use size exclusion chromatography to achieve systematic fractionation of distinct exRNA carrier populations. This approach is clear, reproducible, and highly relevant to a very wide range of EV and liquid biopsy researchers. However, there is a fundamental error in assuming or asserting which fraction represents what specific carrier type since this manuscript's methods only evaluate for HDL lipoproteins (ApoA), not LDL, VLDL, and chylomicron proteins (ApoB proteins; B-100 for VLDL and B-48 for chylomicrons). Size exclusion chromatography resolves only by size, and therefore does not separate large lipoprotein particles (LPPs, including LDL, VLDL, and chylomicrons) and small and medium EVs of the same sizes. VLDL sizes range from 30 – 80 nm, LDL from 20-30 nm and chylomicrons from 75 – 600 nm. Multiple prior publications have demonstrated that in addition to SEC or ultracentrifugation, density gradients or other methods such as density "cushions" are required to separate EVs from similarly sized LPPs in order to conclude that particular carried cargo is EV-associated, rather than LPP-associated. Moreover, there is no evidence regarding where exomeres, supermeres, and other 20-80 nm extracellular particles are found in the bins denoted in figure 1e.
 - a. ApoA Westerns demonstrate fractionation of HDL in later SEC fractions, but does not provide any information about VLDLs, which overlap in size with sEVs and carry ApoB proteins rather than ApoA. It is likely that ApoB / ApoB100 positive VLDL particles are significantly present in both sets of fractions designated as small and medium EVs. In order to determine whether it may be best to reclassify the fraction classification from small and medium EV sets include VLDLs. The authors should perform Westerns or high sensitivity ELISAs for ApoB and ApoB-100 at least to ascertain the degree of lipoprotein carry-over. Since only one copy of the ApoB protein is included in each VLDL, while multiple copies of tetraspanins are present in each EV, 10-100x lower amounts of ApoB detected than tetraspanin by Western would indicate similar numbers of VLDL particles present.
 - b. Immunoblots in 4a should include an unfractionated control sample
 - c. Clear-cut designations assigned in Fig 1e are also not technically accurate, as is evident in the TEM figures shown in 1c. Multiple small (20nm and smaller) structures are evident in the background of fraction 6 consistent with HDL or LDL or other extracellular structures (exomeres, supermeres, etc), and the 20-80nm structures noted in Fractions 6 and 8 could as well be LDL, VLDL, or other extracellular structures as EVs.
 - d. All of the TEM figures shown in Figure 1 should include wider fields of view with zoomed in areas / nested views in order to more completely demonstrate the distribution of particle types that are present in each fraction. (as recommended in MISEV 2018).
 - e. TEM and other QC data and figures should also be provided for at least 2/5 samples for each type of donor (healthy, multiple myeloma, lung cancer, and liver cancer).
 - f. All primary data sets, not just aggregate analysis figures, should be made available for nanoparticle analyses and QC tests.
2. Among the labels used for fraction groups, the labels are not consistent (see page 4, lines 130-135)
3. Stranded SmartSeq was used. Why stranded instead of oligo-dT-based? What evidence led the authors to assume fragmentation of mRNAs in EVs? Please provide this information in the manuscript.

4. Were the reads mapped uniformly along mRNA contigs? How were the NGS read lengths chosen and how were the NGS parameters optimized? Is it equally optimal across fraction groups, ie across categories of carriers?

5. The conclusions on page 8 could be more informative. As noted at line 304, prior studies suggest 1 miRNA per 1-100 EVs, with lower densities expected for longer RNA species. Based on this manuscript, the authors claim 99% of plasma in cf-mRNA is EV associated. What do the authors find in terms of distribution of cf-mRNA copies / EV number? And with what coverage?

6. Donor plasma products vary significantly when variables in diet (time since last meal), exercise, and time of day vary. Were circadian, prandial, gender, age, disease stage, and other donor-to-donor / sample-to-sample variables controlled? Were all lung cancer cases NSCLC? And were all of the NSCLC tumors squamous? Were the lung tumors HPV+? A table including pertinent demographic details (especially disease stage, grade, and specific histology) would be helpful.

Reviewer #2 (Remarks to the Author):

This work presents data supporting the view that small RNAs in blood plasma fractionate on size exclusion chromatography (SEC) with particles that have the bona fide EV membrane marker, CD9. Most of the results on the selectivity of this process and changes associated with small particles isolated by SEC from samples taken from cancer patients suggest that RNase protected RNAs may be of diagnostic value. The claim is made at the outset that these RNAs are in extracellular vesicles. However, SEC alone, in spite of EM detection of what appear to be membrane vesicles in the SEC fraction eluting just after the void volume, is not adequate to distinguish large protein particles or aggregates from bona fide membrane vesicles.

Critical to the claim that particular RNAs are sequestered within vesicles, the authors examined pooled fractions eluted just after the void volume from the SEC column representing what they call medium and small EVs. Sedimented material was enriched in immunoprecipitable CD9 which was not seen in later elution fractions (Fig. 4A). Thus, as expected, EVs do fractionate as expected in the SEC. Next, and critically, the authors sample a subset of RNAs for the loss of RNase protection in incubations containing Triton X-100. The results for 4 of the 6 selected RNAs are convincing but we are not told what RNAs these represent and more importantly, we have no way of knowing if they are representative of the large number of particle-bound RNAs found in normal and cancer patient samples. To this end, a superior approach would have been to fractionate the particles on a buoyant density gradient to separate vesicles from non-membrane particles. An alternative which comes right from this work would be to use the immobilized CD9 antibody to pull down vesicles and then to repeat the analysis on the full range of RNAs in material treated with RNase with or without detergent. Without this further fractionation to affirm the vesicle association of the many RNAs studied in this work, I am less confident of the conclusion the authors wish to draw.

Minor points:

1. The numbering system for the SEC fractions is incompletely described. The figures have two numbering systems for the fractions: The column fractions are labeled 1-40 (Fig. 1B) but then in 1E and beyond we are given numbers ranging from FR14- FR3033. Please explain.

2. The supplemental figures are not numbered nor are they explained with Fig. legends.

Selective enrichment of plasma cell-free messenger RNA in cancer-associated extracellular vesicles

Hyun Ji Kim^{1,2}, Matthew J. Rames^{1,2}, Florian Goncalves^{1,2}, C. Ward Kirschbaum¹, Breeshey Roskams-Hieter¹, Elias Spiliotopoulos¹, Josephine Briand¹, Aaron Doe¹, Joseph Estabrook^{1,3}, Josiah T. Wagner^{1,4}, Emek Demir^{1,3,8}, Gordon Mills^{5,6,7}, Thuy T. M. Ngo^{1,2,5,8*}

Dear Editor,

We would like to thank the reviewers for recognizing the significance and the novelty of our manuscript. The reviewers raised important concerns to be considered. We revised our manuscript to address all the concerns. The changes are made as follows:

- We performed additional experiments to provide the distribution of cell-free RNA length using bioanalyzer analysis and PCR with primers targeting RNA long amplicons of 898 bp and short amplicons of 80 bp located at 5' end, 3' end and the middle of the transcript (**NEW Supplementary Figure S3**).
- Importantly, we performed additional immunoprecipitation experiments using antibodies against canonical EV marker CD9 and lipoprotein marker ApoB followed by western blot (**Revised Figure 4a & Revised Supplementary Figure S8**) and qRT-PCR (**NEW Figure 4b**). Notably, major ApoB peaks were in non-EV particles (FR912) and early protein-enriched fractions (FR1619), while the majority of cf-mRNA was in the earlier fractions. qRT-PCR followed by immunoprecipitation further confirmed that cf-mRNA is enriched in EV-associated particles.
- We provided expanded TEM images including low-magnification FOVs (**NEW Supplementary Figure S1**), raw EV/particle size table (**NEW Supplementary Table S2**), and RNA sequencing coverage (**NEW Supplementary Figure S4**).
- We also updated clinical information to include disease stages and subtypes (**Revised Table S1**).
- We revised our main text and supplementary document to improve clarity, describe the new results, and discuss some limitations.

Our new data and analyses were consistent with our initial findings and further confirmed the main conclusion of the manuscript that the majority of cf-mRNA in plasma was enriched and protected in EVs. We believe that our revised manuscript satisfactorily addressed all the questions raised by the reviewers.

Please our point-by-point response to reviewers' concerns below:

Reviewer #1 (Remarks to the Author):

The manuscript by T.T.M. Ngo and colleagues, "Selective enrichment of plasma cell-free mRNA in cancer-associated extracellular vesicles," presents a highly detailed and informative view of cell free mRNA in plasma. Moreover, the authors identify specific cf-mRNA signatures associated with multiple cancers, including lung cancer, liver cancer, and multiple myeloma. The results of this manuscript

beautifully demonstrate how “dissecting,” or fractionating, the extracellular carriers of mRNA facilitates the identification of certain disease-specific cfRNA signatures. The authors’ approach is distinctive both in its fractionated analysis of extracellular RNA carriers and in the use of a stranded SmartSeq NGS approach to identify cell-free mRNA, along with the use of ERCC spike in controls for exRNA normalization across highly variable samples.

1. The authors use size exclusion chromatography to achieve systematic fractionation of distinct exRNA carrier populations. This approach is clear, reproducible, and highly relevant to a very wide range of EV and liquid biopsy researchers. However, there is a fundamental error in assuming or asserting which fraction represents what specific carrier type since this manuscript’s methods only evaluate for HDL lipoproteins (ApoA), not LDL, VLDL, and chylomicron proteins (ApoB proteins; B-100 for VLDL and B-48 for chylomicrons). Size exclusion chromatography resolves only by size, and therefore does not separate large lipoprotein particles (LPPs, including LDL, VLDL, and chylomicrons) and small and medium EVs of the same sizes. VLDL sizes range from 30 – 80 nm, LDL from 20-30 nm and chylomicrons from 75 – 600 nm. Multiple prior publications have demonstrated that in addition to SEC or ultracentrifugation, density gradients or other methods such as density “cushions” are required to separate EVs from similarly sized LPPs in order to conclude that particular carried cargo is EV-associated, rather than LPP-associated. Moreover, there is no evidence regarding where exomeres, supermeres, and other 20-80 nm extracellular particles are found in the bins denoted in figure 1e.

Response: We sincerely thank the reviewer for giving thoughtful comments about evaluating large lipoprotein particles in SEC fractions. To address these considerations, we have revised our manuscript as follows:

- 1) Main text to include a description of lipoproteins as potential carriers within the introduction, including a contribution of lipoproteins (chylomicron, VLDL, LDL, HDL) in our fractions, and change the fraction labels accordingly to particle and protein enriched (formerly EV and non-EV).
- 2) Additional immunoprecipitation experiment with ApoB followed by western blot to analyze the distribution of ApoB+ lipoproteins across SEC fractions. We observed major ApoB peaks were in non-EV particles (FR912) and early protein-enriched fractions (FR1619), while the majority of cf-mRNA was in the earlier fractions (FR14 and FR58).
- 3) Additional immunoprecipitation experiments with CD9, ApoB, and IgG controls followed by qRT-PCR to examine cf-mRNA contribution from CD9+ EVs or ApoB+ lipoprotein fractions. qRT-PCR analysis revealed that the majority of cf-mRNA recovered from immunoprecipitation was significantly enriched only in CD9-captured EVs compared to apolipoprotein-captured fractions.
- 4) We thank the reviewer for the comments regarding exomeres/supermeres. Although we observed FR912 contains type of particles smaller than 50 nm, we did not detect EV canonical marker, ApoA1, nor Ago2 in this fraction by IP-WB. James W. Clancy et al, defined nanoparticles smaller than 50 nm could potentially be even exomeres and supermeres [1]. While exomeres and supermeres are shown to express Ago2 [1, 2], we did not detect Ago2 in the FR912. Although these smallest sizes are consistent with LDL, more detailed molecular characterization of these particles is beyond the scope of this study and requires further investigation. We discussed the above content in the revised discussion (page 9).

a. ApoA Westerns demonstrate fractionation of HDL in later SEC fractions, but does not provide any information about VLDLs, which overlap in size with sEVs and carry ApoB proteins rather than ApoA. It is likely that ApoB / ApoB100 positive VLDL particles are significantly present in both sets of fractions designated as small and medium EVs. order to determine whether it may be best to reclassify the fraction classification from small and medium EV sets include VLDLs. The authors should perform Westerns or high sensitivity ELISAs for ApoB and ApoB-100 at least to ascertain the degree of lipoprotein carry-over. Since only one copy of the ApoB protein is included in each VLDL, while multiple copies of tetraspanins are present in each EV, 10-100x lower amounts of ApoB detected than tetraspanin by Western would indicate similar numbers of VLDL particles present.

Response:

We agree with the reviewers to include ApoB analysis along with previously performed ApoA to better demonstrate the distribution of these particle types across fractionated plasma. We thank to the reviewer for careful consideration of lipoprotein contribution to observed cf-RNA trends.

- 1) To examine the level of relative lipoprotein carry-over, we performed additional western blots following immunoprecipitation experiments on fractionated plasma using an antibody against ApoB (**Revised Figure 4a and Revised Supplementary Figure S8**). We observed that the non-EV particles (FR912) and (early-protein fractions (FR1619) displayed the highest level of ApoB. The medium and small EV-enriched fractions had a relatively low and moderate levels of ApoB carry-over compared to the protein fractions, respectively. Therefore, as the reviewer suggested, we have addressed EVs and similar size range of lipoprotein particles can be co-eluted and re-defined FR14 and FR58 as large and medium particles instead of medium and small EVs, respectively.
- 2) In addition, we measured the cf-mRNA level by qRT-PCR following immunoprecipitation with canonical EV marker CD9 and lipoprotein marker ApoB within the EV and lipoprotein associated fractions (FR14 and FR58). qRT-PCR results showed that cf-mRNA was significantly enriched in CD9 immunoprecipitates compared to IgG control. However, no significant enrichment in ApoB immunoprecipitates versus IgG control was found. (**NEW Figure 4b**).

Though large lipoprotein particles were co-fractionated with EVs at a detectable level as predicted by the reviewer, the majority of these particles were eluted in later non-EV fractions which did not contain the majority of cf-mRNA. Furthermore, our qRT-PCR of the CD9 and ApoB immunoprecipitation indicated that EVs are the major carriers of cf-mRNA in plasma. We thank the reviewer for the suggestion of these additional IP experiments, which further validated our original conclusions. We revised **Figure 4a and Supplementary Figure S8**, added **NEW Figure 4b**, and reclassified the fractions within the manuscript

b. Immunoblots in 4a should include an unfractionated control sample

Response: The unfractionated plasma control sample was included in the full western images in the **revised Supplementary Figure S8**.

c. Clear-cut designations assigned in Fig 1e are also not technically accurate, as is evident in the TEM figures shown in 1c. Multiple small (20nm and smaller) structures are evident in the background of fraction 6 consistent with HDL or LDL or other extra cellular structures (exomeres, supermeres, etc), and the 20-80nm structures noted in Fractions 6 and 8 could as well be LDL, VLDL, or other extracellular structures as EVs.

Response: We thank the reviewer for this comment. Due to the overlapping size distributions of particles purified by SEC fractions as shown by our TEM data (**NEW Figure 1c and d**), we denoted each fraction by the dominant particle size versus adjacent fractions and reclassified fractions into those containing both EVs and other lipoprotein particles in the result section. Meanwhile our western blot analyses revealed different distributions for: CD9+ EVs in FR14 and FR58, ApoB+ lipoproteins with the highest peak at FR912 and FR1619, and ApoA1 and Ago2 found in later fractions (FR1619, FR2326, and FR3033). We described the degree of lipoprotein carry over and also the possibility of other extracellular structures (exomeres, supermeres, etc.) as some limitations of our study in the discussion. Please see related comment above from question 1, reply #4.

d. All of the TEM figures shown in Figure 1 should include wider fields of view with zoomed in areas/ nested views in order to more completely demonstrate the distribution of particle types that are present in each fraction. (as recommended in MISEV 2018).

Response: Initially, when we performed the TEM image collection, we did not capture the wider fields of view because we focused on zoomed-in view to measure the sizes of the particles. As requested by the reviewer, we reimaged and provided wider fields of view uniformly across previously imaged samples. These are now included in the **NEW Figure 1c and NEW Supplementary Figure S1**. To be consistent with new images shown, particle size analyses were also re-performed on the newly collected data (**New Figure 1d**). New TEM result is consistent with the previous data.

e. TEM and other QC data and figures should also be provided for at least 2/5 samples for each type of donor (healthy, multiple myeloma, lung cancer, and liver cancer).

Response: We performed TEM imaging and other QC data from healthy samples. However, due to the limited availability and amount of blood samples for cancer patients, we were not able to perform additional TEM/QC analysis on samples from cancer patients. We noted TEM and other QC data were tested for healthy individuals in the method section and discussed this limitation in the discussion section. Please note that we applied uniform processing and analysis protocols for all samples.

f. All primary data sets, not just aggregate analysis figures, should be made available for nanoparticle analyses and QC tests.

Response: We provide raw counts of EV sizes measured by TEM in the **NEW Supplementary Table S2**. TEM images of particle diameters were measured manually using Fiji line tool for all observed and distinct particles above ~20nm in diameter.

2. Among the labels used for fraction groups, the labels are not consistent (see page 4, lines 130-135)

Response: To clarify, our fractions are collected every 4 mL which starts from the 0 mL mark. Therefore, the volume it starts always starts 1 mL lower than our abbreviated fraction (i.e. FR14: eluted from 0 – 4mL, etc.). This was important to not confuse the researchers processing the fractionation. Following this logic, TEM/QC fractions were collected in 2mL increments (i.e FR2 was eluted 0-2mL). We have checked for fraction (FR) labeling consistency throughout the manuscript which we hope clarifies any prior confusion.

3. Stranded SmartSeq was used. Why stranded instead of oligo-dT-based? What evidence led the authors to assume fragmentation of mRNAs in EVs? Please provide this information in the manuscript.

Response: We used stranded SMARTseq method to prepare libraries for sequencing to effectively capture short fragments of RNA in plasma. We added a **NEW Supplementary Figure S3** to display the size distribution of plasma cell-free RNA measured by bioanalyzer and PCR. Bioanalyzer analysis showed that RNA purified from plasma are short fragments with predominant peak of less than 200 nt (**NEW Supplementary Figure 3a**). To confirm the fragmentation of plasma cell-free RNA length, we designed PCR primers targeting a RNA long amplicon of 898 bp and short amplicons of 80 bp located at 5' end, 3' end and the middle of the transcript of a tissue specific gene ALB (**NEW Supplementary Figure S3b**). All short amplicons at three locations on the transcript was amplified while the long amplicon is not detected in plasma cell-free RNA.

4. Were the reads mapped uniformly along mRNA contigs? How were the NGS read lengths chosen and how were the NGS parameters optimized? Is it equally optimal across fraction groups, ie across categories of carriers?

Response: The sequencing reads cover entire mRNA coding contigs (**NEW Supplementary Figure S4**). We choose NovaSeq S4 lanes for paired-end read x 150 bp sequencing. We applied uniform processing protocol from RNA purification, library preparation and sequencing parameters for all samples across fractions. All fractionated plasma samples were also randomized to reduce sample batch effect. Since we spiked ERCC RNA control mix into plasma fraction as a control and saw no differences across categories of carriers, NGS read length and parameters are equally optimal across fraction groups.

5. The conclusions on page 8 could be more informative. As noted at line 304, prior studies suggest 1 miRNA per 1-100 Evs, with lower densities expected for longer RNA species. Based on this manuscript, the authors claim 99% of plasma in cf-mRNA is EV associated. What do the authors find in terms of distribution of cf-mRNA copies / EV number? And with what coverage?

Response: We agree with the reviewer that the conclusion could be more informative if we can estimate the distribution of cf-mRNA copies / EV numbers. However, we do not know precisely the extraction efficiency of EV and RNA purification due to the unknown nature of the initial intrinsic cf-mRNA/EV amount in plasma. We noted this limitation in the discussion (page 9)

6. Donor plasma products vary significantly when variables in diet (time since last meal), exercise, and time of day vary. Were circadian, prandial, gender, age, disease stage, and other donor-to-donor / sample-to-sample variables controlled? Were all lung cancer cases NSCLC? And were all of the NSCLC tumors squamous? Were the lung tumors HPV+? A table including pertinent demographic details (especially disease stage, grade, and specific histology) would be helpful.

Response: We agree with the reviewer that plasma products can vary depending on the diet, exercise, circadian clock, prandial, gender, age, disease stage, and other donor-to-donor / sample-to-sample variables. These parameters were not controlled in our experiments. We could like to note, in a previous study, we did not observe statistical differences in the total amount of cfDNA and cfRNA and the level of housekeeping transcript GAPDH for the samples collected from the same individuals through different time of day [3]. In that study, we primarily observed interpersonal variation of circulating biomarkers. In this study, we used 5 biological “replicates” as 5 donors in each disease group within the scope of this study. Further studies are needed to examine the relative effect of these variables on cf-RNA levels, carriers, or associated fluctuations. In our study, sample-to-sample variability is controlled by randomizing samples which were spiked in with the same amount of control ERCC RNA.

For Lung cancer cases, 3/5 patients were adenocarcinomas and 2/5 were squamous cell carcinoma, all were treatment naive, and generally early stage. In order to better interpret patient cancers analyzed, we have provided available clinical patient data and demographic details (especially disease stage, subtype, and treatment status). These clinical information for cancer patients used within this study was added to **revised Supplementary Table S1**.

Reviewer #2 (Remarks to the Author):

This work presents data supporting the view that small RNAs in blood plasma fractionate on size exclusion chromatography (SEC) with particles that have the bona fide EV membrane marker, CD9. Most of the results on the selectivity of this process and changes associated with small particles isolated by SEC from samples taken from cancer patients suggest that RNase protected RNAs may be of diagnostic value. The claim is made at the outset that these RNAs are in extracellular vesicles. However, SEC alone, in spite of EM detection of what appear to be membranes vesicles in the SEC fraction eluting just after the void volume, is not adequate to distinguish large protein particles or aggregates from bona fide membrane vesicles.

Critical to the claim that particular RNAs are sequestered within vesicles, the authors examined pooled fractions eluted just after the void volume from the SEC column representing what they call medium and small Evs. Sedimented material was enriched in immunoprecipitable CD9 which was not seen in later

elution fractions (Fig. 4A). Thus, as expected, EVs do fractionate as expected in the SEC. Next, and critically, the authors sample a subset of RNAs for the loss of RNase protection in incubations containing Triton X-100. The results for 4 of the 6 selected RNAs are convincing but we are not told what RNAs these represent and more importantly, we have no way of knowing if they are representative of the large number of particle-bound RNAs found in normal and cancer patient samples. To this end, a superior approach would have been to fractionate the particles on a buoyant density gradient to separate vesicles from non-membrane particles. An alternative which comes right from this work would be to use the immobilized CD9 antibody to pull down vesicles and then to repeat the analysis on the full range of RNAs in material treated with RNase with or without detergent. Without this further fractionation to affirm the vesicle association of the many RNAs studied in this work, I am less confident of the conclusion the authors wish to draw.

Response: We thank Reviewer 2 for carefully reviewing our manuscript to emphasize our novel findings and to suggest additional critical experiments. The suggestion of using density gradient or immunoprecipitation using CD9 antibody is consistent with the suggestion from Reviewer 1. We followed this suggestion and performed additional immunoprecipitation using antibodies targeting canonical EV marker CD9 and larger lipoprotein particles (ApoB) followed by western blot and qRT-PCR analysis (**Revised Figure 4a and NEW Figure 4b**). We selected 4 primers which encompass a wide variety of cf-mRNAs such as tissue specific genes (i.e. ALB), housekeeping genes (B2M), a back splicing junction (CORO1C), and ribosomal protein gene (RSP6) that were optimized in-house. We described which cf-mRNAs were tested in a revised Method. We found maximum ApoB peaks were in FR912 and FR1619, while the majority of cf-mRNA was found in FR14 and FR58. Most importantly, qRT-PCR result showed that cf-mRNA is dominantly detected in CD9 captured EVs. Together with sequencing and RNase digestion analysis, this data further confirmed that EVs are the major carriers of cf-mRNA in plasma. Taken these together, our finding supports the notion that EVs carry the majority of cf-mRNA in human plasma. Please see our detailed response to the similar questions from Review 1 above.

Minor points:

1. The numbering system for the SEC fractions is incompletely described. The figures have two numbering systems for the fractions: The column fractions are labeled 1-40 (Fig. 1B) but then in 1E and beyond we are given numbers ranging from FR14- FR3033. Please explain.

Response: In figure 1b, we characterized nucleic acids and protein amount across all fraction collected from 0-40 mL after the void volume using absorption measurement. To accurately analyze the particle distribution, we used transmission electron microscopy (performed on FR4 – 12) as shown in **NEW Figure 1c**. FR4-FR12 were collected in 2mL increments, as such, FR4 contained the SEC column fractions between the 2mL and 4mL mark. TEM imaging analysis of particle distribution displayed three main distribution ranges: larger than 100 nm, between 50 – 100 nm, and less than 50 nm. Then, we denoted each fraction by the dominant particle size and reclassified fractions containing both EVs and other lipoprotein particles in the result section. This resulted three particle-associated fractions (FR14: from 0 to 4 mL, FR58: from 4 to 8 mL, and FR912: from 8 to 12 mL) and 3 protein-enriched fractions (FR1619, FR2326, and FR3033 to collect early-, middle-, and late- eluting proteins within broad protein spectrum) for transcriptomic analysis. Therefore, we did not collect all protein-enriched fractions from 12 to 40 mL.

2. The supplemental figures are not numbered nor are they explained with Fig. legends.

Response: In this revised manuscript, all Supplementary Figures are numbered and explained with figure legends.

1. Clancy, J.W., A.C. Boomgarden, and C. D'Souza-Schorey, *Profiling and promise of supermeres*. Nature Cell Biology, 2021. **23**(12): p. 1217-1219.
2. Zhang, Q., et al., *Supermeres are functional extracellular nanoparticles replete with disease biomarkers and therapeutic targets*. Nature Cell Biology, 2021. **23**(12): p. 1240-1254.
3. Wagner, J.T., et al., *Diurnal stability of cell-free DNA and cell-free RNA in human plasma samples*. Scientific Reports, 2020. **10**(1): p. 16456.

REVIEWERS' COMMENTS:

Reviewer #3 (Remarks to the Author):

The authors have done a good job in addressing my main concern, and similarly a principal concern of reviewer #1. Using the CD9 IP protocol, they have now shown that a selected few of the RNAs they previously showed to be resistant to RNase are located in bona fide vesicles (new Fig 4b). I have no other concerns that need attention and feel the work merits publication in your journal.

Randy Schekman